# Diverse prehistoric cattle husbandry strategies in the forests of Central Europe

Rosalind E. Gillis [1,2,3] ✉, Iain P. Kendall[4], Mélanie Roffet-Salque [4], Marco Zanon [5], Alexandra Anders [6], Rose-Marie Arbogast[7], Peter Bogucki [8], Veronika Brychova[9,29], Emmanuelle Casanova [4,10], Erich Classen [11], Piroska Csengeri[12], Lech Czerniak [13], László Domboróczki[14], Denis Fiorillo[3], Detlef Gronenborn [15], Lamys Hachem[16], János Jakucs[17], Michael Ilett [18], Kyra Lyublyanovics [19], Eva Lenneis[20,21], Arkadiusz Marciniak [22], Tibor Marton[17], Krisztián Oross [17], Juraj Pavúk[23], Joachim Pechtl [24], Joanna Pyzel [25], Peter Stadler[19], Harald Stäuble[26], Ivana Vostrovská [27,30], Ivo van Wijk[28], Jean-Denis Vigne[3], Marie Balasse [3] & Richard P. Evershed[4]

During the sixth millennium BCE, the first farmers of Central Europe rapidly expanded across a varied mosaic of forested environments. Such environments would have offered important sources of mineral-rich animal feed and shelter, prompting the question: to what extent did early farmers exploit forests to raise their herds? Here, to resolve this, we have assembled multi-regional datasets, comprising bulk and compound-specific stable isotope values from zooarchaeological remains and pottery, and conducted cross-correlation analyses within a palaeo-environmental framework. Our findings reveal a diversity of pasturing strategies for cattle employed by early farmers, with a notable emphasis on intensive utilization of forests for grazing and seasonal foddering in some regions. This experimentation with forest-based animal feeds by early farmers would have enhanced animal fertility and milk yields for human consumption, concurrently contributing to the expansion of prehistoric farming settlements and the transformation of forest ecosystems. Our study emphasizes the intricate relationship that existed between early farmers and forested landscapes, shedding light on the adaptive dynamics that shaped humans, animals and environments in the past.

Subsistence practices, such as animal husbandry and agriculture, are central components of human cultural activities[1–3], transforming species and ecosystems and fuelling population growth[4]. Deciphering livestock diets in prehistory holds the key to unravelling insights into past human behaviour, specifically the strategies employed for animal feed management in response to local environments. An especially dynamic phase in the evolution of domesticated cattle occurred in Central Europe during the sixth millennium BCE and was associated with the rapid expansion of the 'Linearbandkeramik' (LBK) culture[3,5]. This expansion occurred over a vast geographical area characterized by diverse forested environments with varying densities of cover[6–9]. Natural clearings within the forests occurring within the proximity of rivers and, as a result of lightning strikes and the activities of wild animals, may have been purposefully and deliberately expanded by animal grazing, opening up areas for settlement and agriculture[10,11]. Initially, short-lived slash and burn clearances were proposed as the

**Fig. 1 | A model of LBK cattle herding and diet with reference to stable isotopes. a,** A model of LBK cattle husbandry across the annual seasons, represented by the trees. Oxygen isotopic ratios of drinking water will vary in temperate regions relative to local temperatures, with high values in summer and low in winter. During the seasonal calendar, we may expect animals to be supplied with additional feed from the forest, that is, leafy hay. We assume the autumn slaughter would remove unwanted males and old unproductive females. **b,** The canopy effect on plant $\delta^{13}C$ values ($\delta^{13}C_p$) in temperate environments. **c,** Hypothetical stable isotope values of sequential samples of enamel bioapatite from cattle molars being raised in forested and open environments in different scenarios. Credit: trees, cattle, deer and sheep icons from Vecteezy.com.

main subsistence strategy for early farmers[12], which was dismissed by weed seed analysis indicating intensive manured crop cultivation plots within forests[13]. If forested environments were utilized for crop cultivation, it raises the question of how they were used for animal husbandry. European woodlands have a rich historical precedent of serving dual roles as shelter and animal feed resource (pasture and supplementary feed, referred to as 'leafy hay'[10,14–16]). Nevertheless, the specific role of forests in LBK animal husbandry and the potential adaptations of these practices in response to various forested environments remain unknown.

The LBK farmers settled initially the seasonally flooded marshes of the Carpathian Basin[6] and the forested steppe of eastern Austria (~5545–5360 cal BCE[17]) before expanding into the regions to the north, east and west (~5380–5315 cal BCE[18–21]). Despite the occupation of a variety of landscapes, often these communities are portrayed as largely homogeneous[22,23], particularly in terms of their subsistence economies[21,24,25]. Cattle (*Bos taurus*) herds were of singular importance to the LBK communities, with mortality profiles indicating herds served dual purposes for milk and meat production[5,26,27]. Herd composition probably fluctuated with spring calving and the periodic slaughter of unwanted males and unproductive females, particularly before the onset of winter (Fig. 1a). Seasonal production of milk served as a valuable food source, easily transformed into a range of storable products, including casein balls or hard cheeses, sustaining communities during lean periods and food crises[28,29]. Consistent access to feed is essential for herds as any interruption can lead to nutritional stress, adversely affecting growth, fertility and milk production[30]. Hence, pasture and fodder management are vital components of animal husbandry with the inclusion of supplementary feed, such as leafy hay, playing a crucial role when access to pastures is restricted (Fig. 1a). The latter can also improve animal health and milk let down and quality[16]. Early farmers demonstrated considerable success in adapting

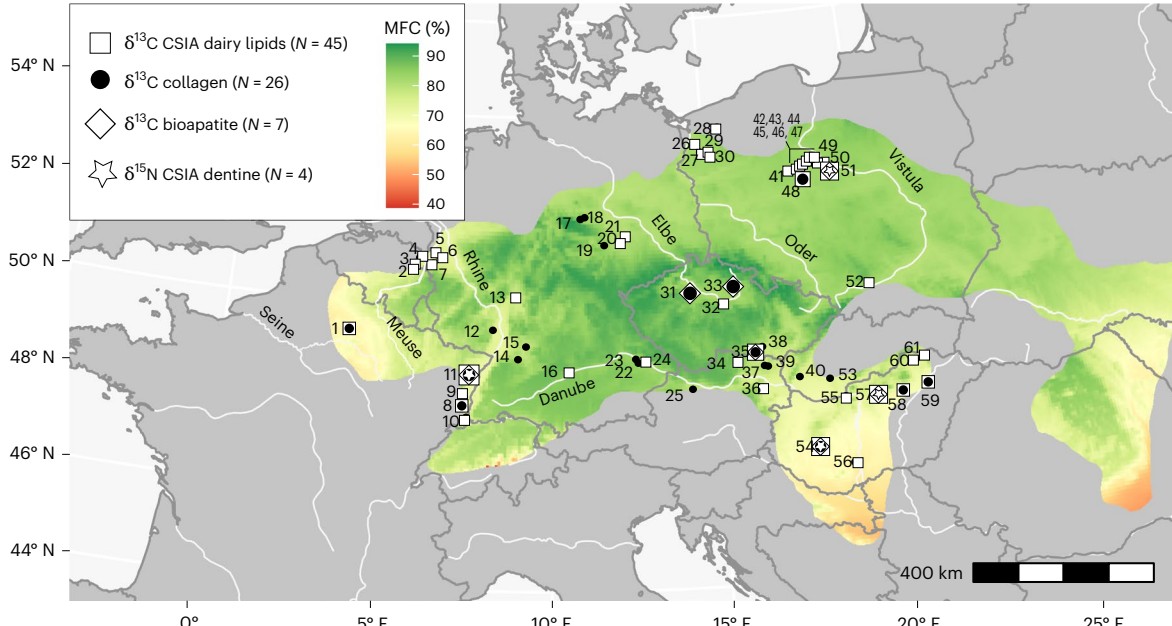

**Fig. 2 | Site location with reference to modelled MFC.** Site locations and MFC calculated from past forest reconstructions[57] within the LBK distribution based on ref. [20] (Methods). All sites are LBK, except for Alföld Linear Pottery (ALP, Supplementary Table 1). *N*, number of sites. (1) Cuiry-lès-Chaudardes, (2) Maastricht-Cannerberg, (3) Maastricht-Klinkers, (4) Geleen-Janskamperveld, (5) Erkelenz-Kückhoven, (6) Konigshoven 14 (FR 5), (7) Langweiler 8 (ref. [48]), (8) Ensisheim-Ratfeld[19,21], (9) Colmar[19], (10) Sierentz[19], (11) Bischoffsheim[19,21,66], (12) Herxheim[54], (13) Kilianstädten, (14) Vaihingen an der Enz[33], (15) Heilbronn-Neckargartach[21], (16) Dillingen-Steinheim, (17) Derenburg Meerenstieg II[53], (18) Halberstadt Sonntagsfeld[53], (19) Karsdorf[53], (20) Altscherbitz, (21) Brodau[48], (22) Aiterhofen[21], (23) Lerchenhaid[21], (24) Stephansposching, (25) Rutzig/Haid[21], (26) Płonia 2, (27) Brzezin 7, (28) Karwowo 1, (29) Żalęcino, (30) Żuków, (31) Černý

Vůl[34,55,66], (32) Bylany[50,51], (33) Chotěbudice[34,55,66], (34) Stroegen, (35) Těšetice-Kyjovice[21,66], (36) Brunn am Gebirge, (37) Gnadendorf[21], (38) Vedrovice-Sídliště[21], (39) Asparn a. d. Zaya/Schletz[21], (40) Blatné[21], (41) Chabsko 24, (42) Żegotki, (43) Bożejewice 22/23, (44) Rożniaty 2, (45) Radojewice 29, (46) Kuczkowo 5, (47) Siniarzewo 1, (48) Kopydłowo 6 (refs. [47,56]), (49) Ludwinowo 7 (refs. [26,27,46,66]), (50) Bodzia 1, (51) Kruszyn 13, (52) Modlnica 5, (53) Vráble-Veľké Lehemby[52], (54) Balatonszárszó-Kis-erdei-dűlő[21,66], (55) Štúrovo, (56) Tolna-Mözs-Községi-Csádés-földek, (57) Apc-Berekalja I[66], (58) Füzesabony-Gubakút (ALP)[21,66], (59) Polgár-Ferenci-hát (ALP)[21], (60) Garadna- Elkerülő út (ALP) and (61) Felsővadász-Várdomb (ALP). The outline of the LBK distribution was adapted from ref. [20] under a Creative Commons licence CC BY 4.0.

cultivation practices in new landscapes[13]. Consequently, it is highly feasible that LBK farmers experimented with domesticated animal diets, where variation in forest environments played a potential role in shaping regional husbandry strategies.

Physical evidence for forest grazing and supplementing animal diets with leafy hay has been identified in waterlogged ruminant dung found at late Neolithic Swiss villages[31]. Without direct archaeo-botanical evidence of fodder, the past use of forest resources can be investigated using stable carbon isotope values of preserved animal tissues[32–37]. Plant $\delta^{13}C$ ($\delta^{13}C_p$) values are governed by the growing conditions and their influence on the efficiency of photosynthesis[38,39]. Under dense forest canopies, plants exhibit low $\delta^{13}C_p$ (ref. [40]) values compared with those in grassland environments[41] (canopy effect; Fig. 1b). This is a combination of the atmospheric $CO_2$ under the canopy being $^{13}C$ depleted due to the decomposition of $^{13}C$-depleted organic matter[41] and the low light levels that reduce photosynthetic efficiency, resulting in plant $^{13}C$ discrimination[38,39]. These values are passed on to consumers, and thus milk and body tissues from forest-based cattle are expected to display relatively low $\delta^{13}C$ values[42]. Previous studies have hinted at a distinct reliance on forest resources in LBK cattle diets, indicated by lower $\delta^{13}C$ values in bone collagen ($\delta^{13}C_{coll}$) compared with those observed in sheep diets[36,37], warranting further investigation. To delve into this aspect, the dual analysis of carbon ($\delta^{13}C_{bioap}$) and oxygen isotopes ($\delta^{18}O_{bioap}$) in incrementally sampled enamel of high-crown ruminant molars is a valuable tool for understanding animal diets at a seasonal scale[43,44] (Fig. 1c). This method leverages the seasonal framework provided by $\delta^{18}O_{bioap}$ to interpret $\delta^{13}C_{bioap}$ values (Methods), providing a means of reconstructing human behaviour and animal feed regimes (Fig. 1c).

Open canopy forest introduce complexity to the narrative since both herbaceous plants and collected leafy hay fodder from these environments would exhibit similar $\delta^{13}C$ values to those from open grasslands[35,38]. To address this challenge, compound-specific isotope analysis (CSIA) of nitrogen within amino acids (AAs) ($\delta^{15}N_{AA}$ values) of dentine is capable of distinguishing diets derived from either woody or herbaceous plants[45] (Methods), providing a means of validating observations made via $\delta^{13}C_{bioap}$ values[46]. Additionally, CSIA-carbon of dairy lipids preserved in pottery is another method, albeit an underutilized proxy, for probing the diets of prehistoric ruminants[47,48]. In contrast to bone collagen, dairy lipids are biosynthesized on a daily basis, offering precise ruminant dietary information[42,49]. An extensive dataset derived from dairy lipids within the LBK culture assembled through recent EU research projects[19,26,27,47,48,50,51] offers a unique opportunity to explore the utilization of forest resources during lactation.

In this study, we have curated an extensive dataset consolidating $\delta^{13}C$ values derived from cattle bone and enamel bioapatite with $\delta^{18}O$ values, complemented by CSIA-$\delta^{15}N$ determinations of dentine amino acids and CSIA-$\delta^{13}C$ of pottery dairy lipids, specifically targeting the $C_{16:0}$ fatty acid. The dataset incorporates both unpublished and published data, comprising a robust set of 2,418 isotopic measurements sourced from 61 early Central European farming sites[19,21,26,27,33,34,46–48,50–56], all unified by the distinctive LBK material culture (Fig. 2 and Supplementary Tables 1–5). These datasets have been cross-correlated with an array of site-specific environmental parameters, including mean forest cover (MFC)[57], palaeoclimate proxies derived from palynological data[58] (Methods) to ascertain the extent to which LBK herders utilized forests for cattle husbandry. The palaeoclimate proxies include mean precipitation in summer and winter (PMW), mean temperature in summer and

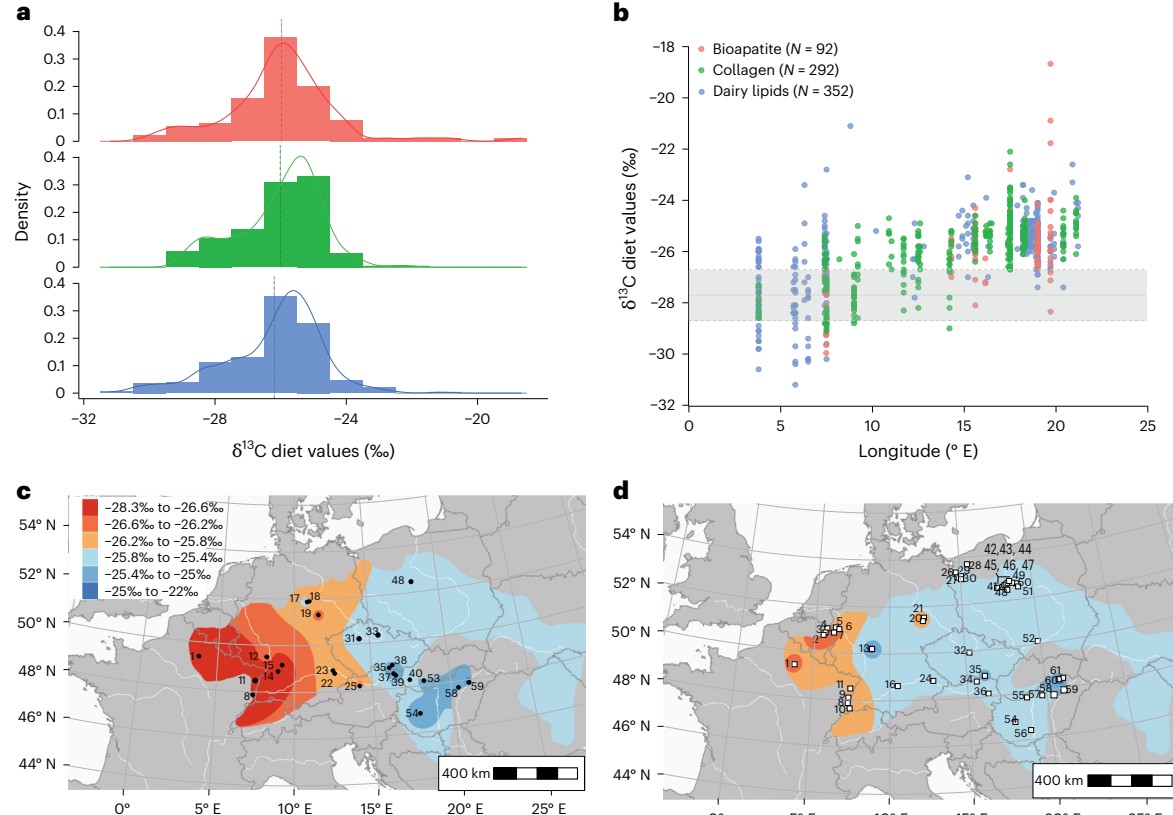

**Fig. 3 | Overview of cattle/ruminant diet δ¹³C values. a,** A histogram of the cattle/ruminant diet $\delta^{13}C$ values ($\delta^{13}C_{diet}$) based on new and published $\delta^{13}C_{16:0}$ values from dairy lipids[19,26,27,47,48,50,51] recovered from pottery vessels (blue, $N = 352$), $\delta^{13}C_{coll}$ values[21,33,52–54,56] (green, $N = 292$) from bone collagen and max/min $\delta^{13}C_{bioap}$ values from the sequential bioapatite samples from cattle molars[34,46,55] (red, $N = 92$) (Supplementary Tables 2–4). **b,** A biplot of longitude and $\delta^{13}C_{diet}$ from the three datasets (colour legend as in **a**). The dotted line with error margins in **a** and **b** represents the diet value of −27.7‰, s.d. of 1‰ ($1\sigma$) for a forest dwelling ruminant animals based on the mean $\delta^{13}C_{coll}$ value from published contemporaneous deer bone collagen data[21,33,53,54] and new data (Supplementary Table 6). **c,** Interpolation map for cattle $\delta^{13}C_{diet}$ inferred from $\delta^{13}C_{coll}$ values (black circles). **d,** Interpolation map for dairy ruminant $\delta^{13}C_{diet}$ inferred from $\delta^{13}C_{dairy}$ values (white squares, colour legend as in **c**). The outline of the LBK distribution was adapted from ref. 20 under a Creative Commons licence CC BY 4.0.

winter, and modern river information, incorporating river drainage systems, Strahler stream order and distance to the nearest river (Supplementary Table 1). Employing a multi-isotopic approach enables detailed exploration of the role of forests in cattle diets and the impact on herds, communities and ecosystems. Through this comprehensive analysis, we aim to shed light on the dynamic interplay between early farming practices, resource utilization patterns and the intricacies of the ecological dynamics that shaped early farming communities.

## Results and discussion

### Stable carbon isotopes

To compare datasets, we calculated the diet $\delta^{13}C$ values ($\delta^{13}C_{diet}$) using established offsets[42,59,60] (Table 1 and Methods). Dietary values ranged from −31.2‰ to −19.8‰, with no significant difference found between three datasets (analysis of variance, d.f. of 2, $P = 0.1$; Fig. 3a). Overall, the $\delta^{13}C$ values fall within the anticipated distribution for cattle feeding within a $C_3$ environment (Fig. 3 and Table 1). Employing river drainage systems to delineate microregions within the dataset facilitated an evaluation of site distribution and preservation conditions. Notably, the upper Rhine, upper/middle Danube, Vistula and Oder drainage areas were well represented, each comprising four sites or more (Supplementary Table 1). Microregions exhibiting suboptimal preservation conditions for bone collagen, such as the Meuse, middle Rhine and Vistula regions, were discernible. The ruminant dairy lipid ($\delta^{13}C_{dairy}$) dataset stands out as the largest, boasting optimal regional distribution, albeit with inherent biases in both distribution and recovery rates.

Notably, the Oder region yielded 61 measurements ($N$ sites 13), while a single site close to the river Aisne within the larger drainage basin of the Seine provided 49 measurements (Cuiry-lès-Chaudardes). A greater intersite variation was observed in $\delta^{13}C_{dairy}$ compared with $\delta^{13}C_{coll}$ values (see, for example, Table 1). In light of slaughter management profiles from cattle[5] and sheep/goat[61], we assume that these dairy lipids originate from cattle, which is supported by the lack of difference between $\delta^{13}C_{diet}$ values derived from bone collagen and dairy lipid $\delta^{13}C$ values within individual sites. Furthermore, a previous large-scale analysis has demonstrated that sheep/goat and cattle were subject to different dietary regimes[36]. We recognise that this does not rule out sheep/goat being exploited for milk and fed on the same diet during lactation. Animal teeth are generally well preserved, but their recovery is contingent upon depositional and excavation practices, which limited the availability of suitable cattle teeth to seven sites. These, however, were representative of the regions studied.

Early farmers of Central Europe occupied a succession of different forest types: marshland and gallery (Carpathian Basin[6]), open deciduous (Polish lowlands[56]) and forest steppe (Czechia[7,9]) to more closed mixed oak forests of the Rhine Valley[8]. This is partly reflected in the MFC values, which range from 85% in Poland, Czechia, Slovakia and Southern Germany to 60% in the Paris basin (Fig. 2). In regions characterized by high MFC, the anticipation was a greater dependence on forest resources given the reduction in available open pastures and the consumption of plants with lower $\delta^{13}C$ values due to the 'canopy effect'[38]. It should be noted that MFC is not a proxy for woodland

**Table 1 | Statistical summary (average, range and average s.d.) of δ¹³C values (N = 1,521) determined for bone collagen (δ¹³C$_{coll}$), dairy lipids (δ¹³C$_{dairy}$) and bioapatite (δ¹³C$_{bioap}$) (Supplementary Tables 2–4)**

| | δ¹³C$_{coll}$ (N = 292, N sites 45) | | δ¹³C$_{dairy}$ (N = 352, N sites 23) | | δ¹³C$_{bioap}$ (N teeth 46, N sites 7, N = 877) | |
| --- | --- | --- | --- | --- | --- | --- |
| | Raw (‰) | Diet (−5.1‰)[60] | Raw (‰) | Diet (1.5‰)[42] | Raw (‰) | Diet (−14.5‰)[59] |
| Average | −20.9 | −26 | −27.7 | −26.2 | −11.5 | −26.1 |
| Range | −24 to −17 | −29.1 to −22.1 | −32.6 to −22.6 | −31.2, −21.1 | −15.5 to −4.2 | −30.1 to −18.8 |
| Average s.d. | 1.3 | 1.3 | 1.5 | 1.5 | 1.5 | 1.5 |

The raw values were used to derive diet values using a known enrichment factor[42,59,60].

composition, and thus its canopy structure and density[62]; however, the only meaningful significant negative correlation was observed between MFC and δ¹³C$_{coll}$ values (Pearson's correlation, $r = −0.29$, $P = 4.6 × 10^{−7}$; Supplementary Tables 7 and 8). Contrary to expectations, a signifcant positive correlation, that is, where stable isotopic values increase with MFC, was observed between MFC and δ¹³C$_{dairy}$ values (Pearson's correlation, $r = 0.5$, $P < 2.2 × 10^{−16}$; Supplementary Tables 7 and 8). This unexpected finding implies that forested areas were utilized for cattle rearing even in regions where open pastures were available. The challenge of equifinality introduces a constraint on the certainty of categorizing animal diets into specific ecological niches. For instance, plants thriving in waterlogged environments may exhibit δ¹³C values similar to those growing under dense canopies[38,63]. Within our dataset, the average distance between sites and water sources was 1.5 km and MFC values decrease in the vicinity of rivers as evident for areas such as the upper Rhine and Marne[57] (Fig. 2). Consequently, the positive correlation between MFC and dairy lipid δ¹³C values could indicate cattle grazing in waterlogged environments[38,63]. However, this does not rule out the use of forested areas for pasture where open pasture existed.

The δ¹³C$_{coll}$ values of forest-based wild herbivores can provide a baseline for assessing forest use in relation to domesticated herbivores. Therefore, we compared our dataset with an average δ¹³C$_{diet}$ value based on published[21,33,53,54] and unpublished red and roe deer bone collagen from LBK sites (−27.7‰, s.d. of 1‰ (1σ), N = 35; Supplementary Table 6). Strikingly, δ¹³C$_{diet}$ values from sites in the west fell within or below the forest herbivore baseline (Fig. 3b). This is further highlighted in the interpolated maps of δ¹³C$_{diet}$ values based on bone collagen (Fig. 3c) and dairy lipids (Fig. 3d), which become progressively more depleted in ¹³C moving east to west. While our dataset of diet δ¹³C values signifcantly correlated with longitude (Pearson's correlation $r = 0.66$, $P < 2.2 × 10^{−16}$; Fig. 3b and Supplementary Tables 7 and 8), there was no significant correlation between deer values and longitude and MFC[36], which is perhaps a reflection of the small size of the deer dataset. The question remains whether the relationship between cattle diet δ¹³C values and longitude is a reflection of regional adaption of pasturing strategies or the influence of external variables, such as climate, on pasture δ¹³C sources.

### Seasonal diet

Seasonal use of supplementary feed such as leafy hay was explored by examination of the variations in δ¹³C$_{bioap}$ values from teeth of individual animals. The 877 δ¹³C$_{bioap}$ measurements ranged from −15.5‰ to −4.2‰ (Table 1 and Supplementary Table 4), with an average of −11.5‰. However, there were exceptions, notably two individuals from Apc-Berekalja displayed values above this range. While all individuals from Bischoffsheim and one animal from Chotěbudice[34] (CHO09) and Apc-Berekalja I (APC2) had values below −13.2‰, that is, the forest reference (Fig. 4a). These values coincide with low δ¹⁸O values (Fig. 4a and Extended Data Fig. 1), suggesting that animals fed on plants depleted in ¹³C during winter. Within individual teeth, the amplitude between the highest and lowest δ¹³C$_{bioap}$ values ranged from 0.3‰ to 3.5‰ (Fig. 4b), with individuals from Bischoffsheim, Apc-Berekalja I and Balatonszárszó

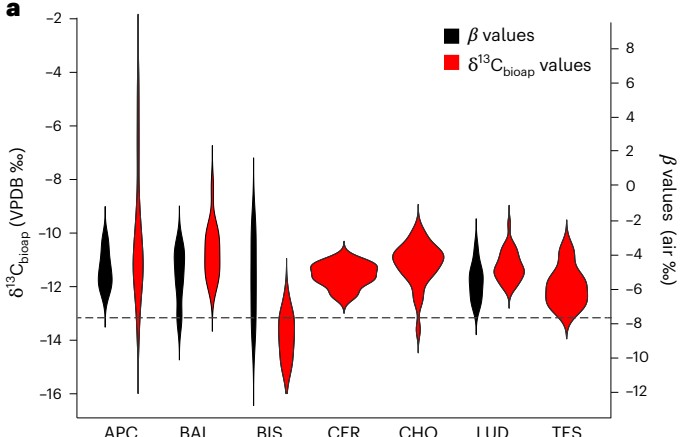

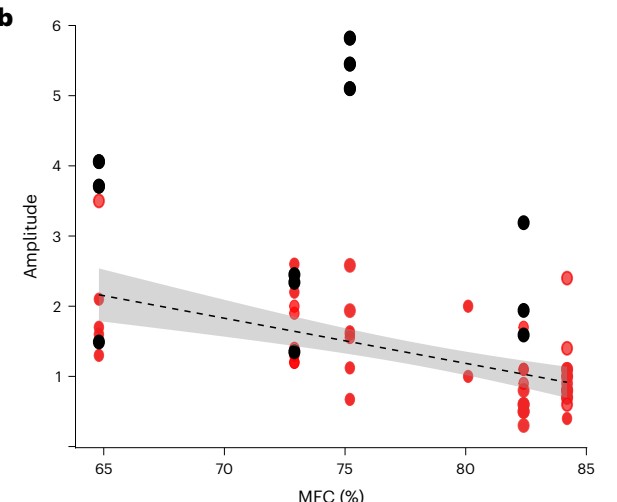

**Fig. 4 | Overview of δ¹³C$_{bioap}$ and dietary $β$ values from cattle molars. a**, Violin plots of δ¹³C$_{bioap}$ values (red) from Apc-Berekalja I (APC), Balatonszárszó (BAL), Bischoffsheim (BIS), Chotěbudice (CHO)[34,55], Černý Vůl[34] (CER), Ludwinowo 7 (LUD)[46] and Těšetice-Kyjovice (TES), and dietary $β$ values (Δ¹⁵N$_{Glx−Phe}$, black) from APC, BAL, BIS and LUD. The dashed black line is the hypothetical δ¹³C$_{bioap}$ value for forest-dwelling herbivore (−13.2‰, based on −27.7‰ adjusted for Δ$_{bioap-diet}$ by −14.5‰ (ref. 59)) and dietary $β$ values for woody plants (−7.7‰ (ref. 45)). **b**, The amplitude (max−min) in δ¹³C$_{bioap}$ values (red) and dietary $β$ values (black) from cattle molars in comparison with MFC (%). The linear regression line is the correlation between δ¹³C$_{bioap}$ and MFC values (Pearson's correlation, $r = −0.7$, $P = 3.84 × 10^{−6}$) with a 95% confidence interval.

exhibiting a wide variation in δ¹³C values of consumed plants. This was significantly correlated with MFC (Pearson's correlation, $r = −0.7$, $P = 3.8 × 10^{−6}$) but not with PMW ($r = 0.32$, $P = 0.03$). These results suggest either diverse dietary plant sources, that is, forests, grassland and/or high seasonal variability in δ¹³C$_p$ values within local pastures due local environmental conditions.

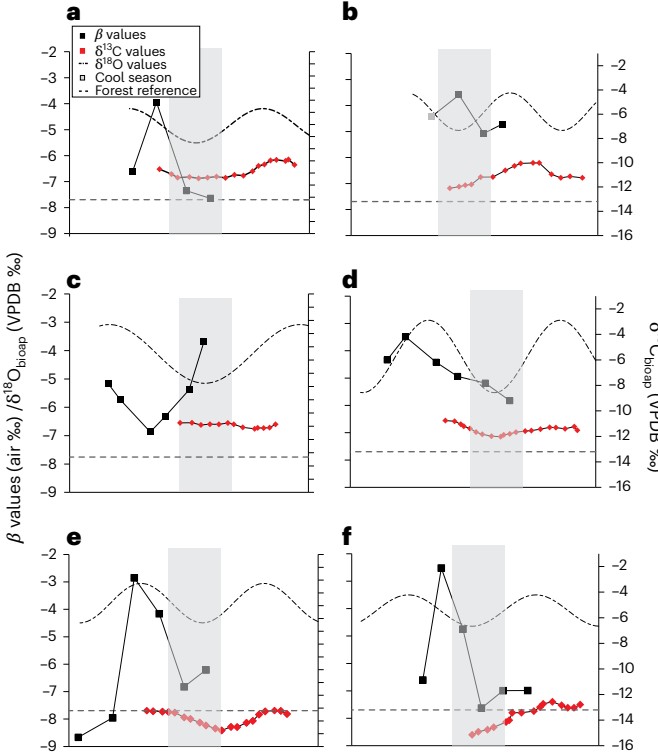

**Fig. 5 | Integrated bioapatite δ¹³C, δ¹⁸O and dentine β values from cattle molars. a–f,** Integrated bioapatite δ¹³C (red diamonds), δ¹⁸O (dashed line) and dentine β values (black squares) from cattle molars from BAL3 (**a**), BAL5 (**b**), LUD1 (**c**), APC1 (**d**), BIS3 (**e**) and BIS7 (**f**). The β values have been adjusted by an estimated 6 months to account for the delay between mineralization of dentine and enamel. The dotted light-grey line is the upper limit of both dietary β values for woody plants (−7.7‰ (ref. 45)) and hypothetical δ¹³C value for forest dwelling herbivore (−13.2‰, based on −27.7‰ adjusted for Δ_bioap-diet by −14.5‰ (ref. 59)). The δ¹⁸O (dashed line) curve is based on modelled values. The light-grey boxes are an approximation of the cold season based on δ¹⁸O values. The x-axes increase in time from left to right, reflecting the development of the tooth over time.

We examined the types of plants consumed using β values based on CSIA-AA δ¹⁵N values of sequential dentine samples of 12 individual molars previously analysed for δ¹⁸O/δ¹³C. The β values can distinguish between different dietary plant sources that is herbaceous (−5.4‰, s.d. of 2.1‰ (1σ)) or woody (−9.3‰ s.d. of 1.6‰ (1σ))[45] (Methods). Woody plant consumption was evident in all individuals from Bischoffsheim while individuals from Balatonszárszó (BAL2/3; Fig. 4a) were borderline. Notably, there was no significant correlation between MFC and β values (Spearman rank correlation, S = 335.4, rho of −0.2, P = 0.6), indicating that greater forest cover does not necessarily preclude the presence of leafy hay in cattle diets (Fig. 4b). Furthermore, at Apc-Berekalja, despite observed low δ¹³C_dairy and δ¹³C_bioap values (APC2), there was no sign of woody plant consumption. Instead, these depleted δ¹³C_bioap values may be indicative of animals grazing in marshy environments[63] typical of the Carpathian Basin or grazing on herbaceous plants growing under a dense forest canopy.

To delve into the seasonality of different plant types within cattle diets, we amalgamated the bioapatite (δ¹⁸O/δ¹³C) and dentine β values from six individuals (Fig. 5a–f and Extended Data Fig. 2), specifically chosen for having more than four β values per tooth. The position of the β values in relation to tooth growth was adjusted to account for the approximate 6 month delay between the mineralization of dentine and enamel[44], while the δ¹⁸O values are modelled based on the approach outlined by Balasse et al.[64]. In individuals from Bischoffsheim, the lowest δ¹⁸O and β values coincide, indicating the consumption of leafy hay during winter, confirming what was suggested by the δ¹³C values

(Fig. 5e,f). BAL3 also displays low β values concurring with low δ¹⁸O values (Fig. 5a) and δ¹³C values, albeit borderline with respect to the woody plant/forest baseline. In contrast, individuals Ludwinowo 1 (LUD1), APC1 and BAL5 exhibit β values indicative of a diet primarily consisting of herbaceous plants (Fig. 5b–d). These findings point to considerable diversity in seasonal dietary regimes in the management of LBK cattle herds.

The findings from Bischoffsheim reveal an early instance of winter foddering involving leafy hay and may explain the notable variation in δ¹³C_dairy values observed at Bischoffsheim (BIS) (1.8‰) and Cuiry-lès-Chaudardes (1.2‰). The large variation in δ¹³C_dairy values observed in Cuiry-lès-Chaudardes may be explained as a reflection of changes in the canopy density during the lactation season (Fig. 1a), as well as mixing of milk from both sheep and cattle. The practice of supplementing winter diets with nutrient-rich feed has direct implications for enhancing female cattle health and fertility, leading to an increase in out-of-season births and improved infant survival rates[65]. The accessibility to leafy hay emerges as a potential factor contributing to the observed higher frequency of out-of-season births in the LBK cattle herds, setting them apart from those managed by other early Neolithic cultures[66]. The occurrence of out-of-season births would have extended the availability of milk for human populations due to overlapping lactation periods. Milk held a pivotal role in the diets of the LBK communities[29], and analogous to traditional cattle land races, LBK cattle probably necessitated the presence of the calf and/or the provision of feed, such as leafy hay, to encourage optimal milk let down[16]. Fodder, particularly leafy hay from specific tree species, is known to stimulate milk let down and it is associated with improved milk quality[16,67]. Within our study, the most abundant dairy fat signals, as seen at Cuiry-lès-Chaudardes, may reflect efficacious use of forest animal feed to increase dairy production.

## Impact of climate on stable isotope proxies

LBK communities occupied two main climatic zones: humid continental and oceanic temperate climate. Humidity and temperature can have a substantial impacts on δ¹³C_p values[38], which are subsequently passed onto grazing herbivores. Exploring the correlation between palaeo-environmental proxies and δ¹³C_diet values revealed that winter climate proxies (PMW and mean temperature in winter), held the most substantial relationship (Supplementary Tables 7 and 8 and Extended Data Fig. 3). In comparison, previous analyses employing modern climatic values found significant correlation between δ¹³C values of cattle bone collagen and summer humidity[37]. Our palaeoclimatic proxies, derived from pollen core data, offer a more accurate representation of LBK climate, albeit averaged over a period marked by precipitation fluctuations[68]. Therefore, it is crucial to note that the observed relationship here between winter climate proxies and δ¹³C values may have not been constant over time. Future detailed examination of the interplay between paleoclimatic conditions and dietary isotopic values could provide valuable insights into the adaptive strategies employed by LBK herding communities, such as seasonal use of supplementary fodder. Such a nuanced understanding would add depth to how the ecological context surrounding early agricultural practices impacted livestock and human societies.

## Conclusions

Foddering and pasturing practices are critical components in the growth and production of domesticated animals. The comprehensive analysis of δ¹³C values from cattle teeth and bone, along with the examination of ruminant dairy lipids coupled with δ¹⁸O_bioap and CSIA-AA δ¹⁵N values of dentine, offers a unique perspective on early cattle herding strategies employed by herders in diverse local forested environments. The detailed exploration underscores the complexity of dietary patterns in cattle herds, challenging conventional expectations based on δ¹³C values from a single tissue alone. Our study reveals distinctive

regional and seasonal variations in cattle pasturing and foddering practices across the LBK cultural zone, notably the use of winter fodder collected from forests in the northwest regions. It is conceivable that forests served as pastures during periods when open areas became inaccessible due to flooding or snow cover. This adaptive utilization of forested spaces highlights the dynamic and resourceful strategies employed by early farming communities to address seasonal challenges and ensure sustainable livestock practices.

The uptake of supplementary fodders may have been a consequence of climate instability during the LBK period. While acknowledging the necessity of refining the temporal and spatial resolution of paleoclimate proxies, our findings underscore the remarkable diversity in herding strategies within seemingly homogeneous archaeological cultures. The study contributes valuable insights to the evolving understanding of species-specific management strategies employed by early farmers, where forests played a crucial role in cattle diets. Moreover, our results lend weight to the idea that LBK herders experimented with domesticated animal diets through observations of improved production connected to supplying nutrient-rich hay. The emphasis on collecting and storing fodder for herds highlights the centrality of cattle to the early farming communities of Central Europe. Provisioning leafy hay not only improved female cattle health, increased out-of-season births and enhanced young calf survival, but also led to an augmented availability of milk. This surplus of dairy products would have probably supported communities during periods of poor harvest and food scarcity.

The arrival of the first farmers and their herds in Central Europe marked an important stage in the evolution of human-modified forested ecosystems. The gradual intensification of woodland usage and management had considerable consequences, particularly the transformation of forested environments. Herding within forests would have played an active role in expanding settlement areas, as grazing animals contributed to the removal of young saplings and undergrowth. The collection of leafy hay alongside other activities may have increased exchanges and encounters between farmers and hunter gatherers, while herding would have contributed to the gradually erosion of forested environments. Our study contributes to the foundation for future research aiming to reconstruct and model the impact of early herding activities on forested environments. By unlocking information about past human behaviours associated with animal pasture and fodder management strategies, we can begin to assess the potential impact of local ecologies and climates on shaping cattle diets during this pivotal period in Central European prehistory.

## Methods

### Lipid residue analysis of pottery vessels and determination of $\delta^{13}C$ values from dairy lipids

Lipid residue analyses and interpretations were based on established protocols[69]. Briefly, 1–3 g samples were taken from potsherds, and their surfaces were cleaned with a modelling drill to remove exogenous lipids (for example, soil or finger lipids arising from handling). The sherds were ground to a powder in a glass pestle with a mortar. The powdered sherd was transferred to a glass culture tube, an internal standard was added for quantification (n-tetratriacontane, 20 µg) and acidified methanol solution ($H_2SO_4$/MeOH, 4% v/v, 5 ml at 70 °C for 1 h) was added. The lipids were then extracted from the aqueous phase with n-hexane (4× 3 ml). The solvent was evaporated under a gentle stream of nitrogen to obtain the total lipid extract (TLE). Aliquots of the TLE were trimethylsilylated using N,O-bis(trimethylsilyl)trifluoroacetamide containing 1% trimethylsilyl chloride (20 µl at 70 °C for 1 h) and redissolved in n-hexane for analysis by gas chromatography (GC) and GC combustion–isotope ratio mass spectrometry (GC-C–IRMS).

All GC analyses were performed on a Hewlett Packard 5890 series II chromatograph. Helium was used as the carrier gas at a constant flow rate (2 ml min⁻¹), and a flame ionization detector was used to monitor column effluent. Trimethylsilylated TLEs (1 µl) were injected through an on-column injector, in track-oven mode onto a fused silica capillary column (50 m × 0.32 mm internal diameter) coated with a dimethylpolysiloxane stationary phase (J&W Scientific, CP-Sil 5 CB, 0.1 µm film thickness). The oven temperature was programmed, after an isothermal hold at 50 °C for 2 min, to 300 °C at 10 °C min⁻¹, followed by a second isothermal hold at 300 °C for 10 min. Peaks were identified by comparison of retention times with those of an external standard and quantification was achieved by the internal standard method. Data acquisition and processing were carried out by the Clarity software.

GC–mass spectrometry (MS) analyses of trimethylsilylated aliquots were performed using a Finnigan Trace MS quadrupole MS coupled to a trace GC. Diluted samples were introduced using a programmable temperature vaporization injector in the splitless mode onto a 50 m × 0.32 mm internal diameter fused silica capillary column coated with a HP-1 stationary phase (100% polymethylpolysiloxane, 0.17 µm film thickness; Agilent Technologies). The initial injection port temperature was 50 °C with an evaporation phase of 1 min, followed by a transfer phase from 50 °C to 300 °C at 14.5 °C s⁻¹, followed by an isothermal hold at 300 °C. The GC oven temperature was programmed as for the GC analyses. The MS was operated in the electron ionization mode (70 eV) with a GC interface temperature of 300 °C and a source temperature of 200 °C. The emission current was 150 µA, and the MS was set to acquire in the range of m/z 50–650 Daltons at 8.3 scans per second. Data acquisition and processing were carried out using the XCalibur 1.2. software. Peaks were identified based on their mass spectra, GC retention times and by comparison with the National Institute of Standards and Technology mass spectral library (v.2.0a).

Compound-specific $\delta^{13}C$ values of fatty acids were determined using an Isoprime 100 GC-C–IRMS system. The same GC conditions were used as for the GC analyses (HP-1 column, 100% dimethylpolysiloxane, 50 m × 0.32 mm × 0.17 µm, Agilent Technologies). Each sample was run at least in duplicate. Instrument stability was monitored by running a fatty acid methyl ester standard mixture every two or four runs. Results were calibrated against a $CO_2$ reference gas injected directly in the ion source as two pulses at the beginning of each run. Instrumental precision was 0.3‰.

Animal fats were identified as dairy lipids when their $\Delta^{13}C$ ($\delta^{13}C_{18:0} - \delta^{13}C_{16:0}$) values were ≤−3.1‰, as proposed by Dunne et al.[70]. We examined $\delta^{13}C$ values of $C_{16:0}$ fatty acids from a total of 352 extracts identified as originating from animal dairy lipids (of which 135 are published[19,26,27,47,48,50,51]) from 44 sites. The species-specific identification of dairy species (cattle, sheep and goats) is not obtainable through the molecular or isotopic composition of the extracts and thus the dairy lipids from this study can come from any of these species. The $C_{16:0}$ fatty acid was chosen over the $C_{18:0}$ fatty acid in this study as $\delta^{13}C_{16:0}$ values reflect the ruminant's diet, while $\delta^{13}C_{18:0}$ values also reflect tissue type[71].

### Bioapatite sampling and stable isotope analysis of cattle molars

Cattle third molars (M3) were selected for stable isotopic analysis (1) because the archaeozoological material was highly fragmented, making it difficult to distinguish between M1 and M2 and (2) to avoid effects from suckling and weaning. Each tooth sampled represents an individual. A minimum of eight M3s with early stages of occlusal wear were sampled where possible. Tooth surfaces were cleaned using an abrasive tungsten drill bit to remove dental calculus, cementum and sediments. Enamel samples were removed by drilling with a diamond bit on the buccal side of the proximal lobe perpendicular to the crown growth axis. The teeth were pre-treated following Balasse et al.[43] omitting the bleach step. Purified enamel samples weighing between 551 and 650 µg were analysed on a Kiel IV device interfaced to a Delta V Advantage IRMS at the Service de Spectrométrie de Masse Isotopique du Muséum national d'histoire naturelle, Paris. The accuracy and precision of the measurements were verified using an internal laboratory calcium carbonate standard (Marbre LM normalized to National Bureau

of Standards 19). Over the period of analysis, an average of eight Marbre LM samples were analysed per run of 38 samples. These gave a mean $\delta^{13}C$ value of 2.13‰, s.d. 0.03‰ ($1\sigma$) (theoretical value normalized to National Bureau of Standards 19 of 2.13‰) and a mean $\delta^{18}O$ value of −1.66‰, s.d. 0.15‰ ($1\sigma$) (theoretical value of −1.83‰). The results are expressed relative to the Vienna Pee Dee Belemnite standard.

### Estimation of animal diets based on $\delta^{13}C$ values

Diet values were calculated using the following enrichment values: $\Delta_{lipids-diet}$ is +1.5‰ based on $\Delta_{lipids-collagen} = 6.6‰$ (ref. [42]) and $\Delta_{collagen-diet} = -5.1‰$ (ref. [60]); $\Delta_{bioap-diet} = -14.5‰$ (ref. [59]). The spacing between diet-inferred $\delta^{13}C$ values and herbivore bone collagen $\delta^{13}C$ values has been proposed to be between 5.1‰ and 5.3‰ (ref. [60]). Here, we use 5.1‰ so as to be comparable to previous stable isotope studies of LBK faunal material[34]. The enrichment of bioapatite in $^{13}C$ varies between species depending on the difference in physiology and size of the species[59]. We have used an enrichment factor of 14.5‰ based on a recent synthesis of the spacing between diet, $CO_2$ breath and bioapatite in animals of different digestive systems[59]. The spacing between collagen and fat $\delta^{13}C$ values has been proposed to be −6.6‰ for consumers of terrestrial $C_3$ diets[42]. The diet-inferred $\delta^{13}C$ values are thus calculated by adding 1.5‰ to the $\delta^{13}C$ values of the $C_{16:0}$ fatty acid.

### CSIA-AAs of cattle molars

Cattle third molar (M3) teeth were sequentially sampled for dentine using the windows created during the bioapatite sampling at six points along the growth axis of each tooth. Dentine was collected as a powder, using a modelling drill with a diamond abrasive drill bit. Once formed, dentine in teeth is not remodelled and therefore the collagen preserves the isotopic composition of the period of formation. For each sample, the AA norleucine was added as an internal standard to ca. 15 mg of dentine. Demineralization of the inorganic fraction and hydrolysis of the collagen was achieved in one step by heating with acid (6 M HCl, 5 ml at 100 °C for 24 h), and the solution was blown to dryness under nitrogen. AA purification and derivatization to N-acetyl isopropyl ester derivatives were prepared according to established protocols[72,73].

AAs were identified by GC–flame ionization detection by comparison with AA standards and quantified by comparison with a known amount of norleucine internal standard. Their $\delta^{15}N$ values were determined by GC-C–IRMS as described in ref. [73] with a modified GC method, using DB-35 capillary column (30 m × 0.32 mm internal diameter; 0.5 μm film thickness; Agilent Technologies), and the oven temperature of the GC was held at 40 °C for 5 min before programming at 15 °C min⁻¹ to 120 °C, then 3 °C min⁻¹ to 180 °C, then 1.5 °C min⁻¹ to 210 °C and finally 5 °C min⁻¹ to 270 °C and held for 1 min. A Nafion drier removed water and a cryogenic trap removed $CO_2$ from the oxidized and reduced sample. Isotopic compositions are expressed using the delta scale as follows: $\delta^{15}N = R_{sample}/R_{standard} - 1$, where $R$ is the $^{15}N/^{14}N$ ratio, and the standard is atmospheric $N_2$ (air). All $\delta^{15}N$ values are reported relative to reference $N_2$ of known isotopic composition, introduced directly into the ion source in four pulses at the start and end of each run. Each reported $\delta^{15}N$ value is the mean of triplicate determinations. A standard mixture of AAs of known $\delta^{15}N$ values was analysed every three runs to ensure acceptable instrument performance standards were accepted if within 1‰ of their known $\delta^{15}N$ values.

Direct evidence of the type of plants (woody or herbaceous) consumed can be determined using the dietary $\beta$ values based on $\delta^{15}N$ CSIA of AA from incremental samples of dentine from cattle molars. These values represent the $\Delta^{15}N_{Glx-Phe}$ values of the plants at the base of the food web (that is, the difference between the $\delta^{15}N$ values of glutamate and phenylalanine), using a known trophic offset of −4.0‰ between cattle and their diet[74]. The dietary $\beta$ values can then be compared with established ranges of $\Delta^{15}N_{Glx-Phe}$ values expected for herbaceous (−5.4‰, s.d. of 2.1‰ ($1\sigma$)) and woody plants (−9.3‰, s.d. of 1.6‰ ($1\sigma$)), based on modern references[45]. This difference in values is probably

due to the involvement of Phe in the phenylpropanoid pathway, by which lignin is produced, leading to isotopic fractionation and enrichment of the remaining Phe pool available for protein biosynthesis. This results in the more negative $\Delta^{15}N_{Glx-Phe}$ values observed in woody plants relative to herbaceous plants, as the former are assumed to produce more lignin.

### Palaeo-environment variables

The identification of past forest composition is hampered by the location of pollen cores as well as modelling uncertainties. Localized exploitation of forest resources may be underrepresented in traditional palaeo-ecological investigations owing to difficulties in capturing small-scale landscape dynamics. These difficulties may variously stem from a lack of targeted investigations or from an absence of suitable archives. To sidestep these issues, and to proceed with a complete comparison of faunal and land cover data, we make use of interpolated reconstructions covering the whole study area. Such large-scale interpolated reconstructions may still be unable to fully resolve local dynamics, yet their use allows us to initiate a comparison between geographically spread-out datasets. The MFC data were generated from the interpolated Holocene reconstructions by Zanon et al.[57]. Here, the authors used the modern analogue technique, where a calibration dataset is built by coupling modern pollen samples with the corresponding satellite-based forest cover data. Reconstruction of past forests are carried out by assigning to every fossil sample the average forest cover of its closest modern analogues. We chose to use MFC values (%) sampled from the 7500, 7250 and 7000 cal BP (that is, 5550, 5300 and 5050 cal BCE) time slices at the location of every site in the faunal dataset and subsequently averaged.

Palaeoclimate information (summer and winter temperature and precipitation) is based on the modelled values presented in ref. [58] and available for the time slice ~7100 ± 100 years cal BP (ca. 5150 ± 100 BCE). We applied inverse distance-weighted interpolation to all data points using the R package gstat 2.0-6 (ref. [75]). The optimal power value for each variable was selected via leave-one-out cross-validation, using the root mean square error as a metric to assess the model performance. We then sampled the interpolated climate values at the location of every site within the data set.

### Statistical analysis

The interpolated diet-inferred value maps (Fig. 3c,d) were produced as follows: the median $\delta^{13}C$ values for each site were interpolated via inverse distance-weighted interpolation through the R package gstat 2.0-6 (ref. [75]). The optimal power value for each variable was selected via leave-one-out cross-validation, using the root mean square error as a metric to assess the model performance. The size of the 'bullseye' depends partly on purely graphical choices (number and width of the colour intervals) and partly on the parameters of the interpolation algorithm. The interpolation procedure was built upon R code available at project SOGA-R's website[76].

Pearson and Spearman Rho correlation tests were performed using the using the free platform R program[77] with the corrplot package[78] used for Extended Data Fig. 3 (Supplementary Table 7). Uncorrected $P$ values from Pearson correlation were compared with those corrected using Bonferroni correction and further assessed with the false discovery rate (Supplementary Table 8). This comparison showed that the correlation results are reliable.

Comparison of $\beta$ values from CSIA-AA $\delta^{15}N$ with $\delta^{13}C_{bioap}/\delta^{18}O_{bioap}$ values (Fig. 5) was calculated as follows: On the basis of the principal that the delay between the dentine crown and mineralization of the crown is around a year, with each process taking around 12–18 months to complete[44], and dentine crown of the third molar in cattle begins development around 6 months (ref. [79]), we created a relative scale for both $\beta$ values from CSIA-AA $\delta^{15}N$ and $\delta^{13}C_{bioap}/\delta^{18}O_{bioap}$ values by first calculating the proportion of the tooth where either the bioapatite or

dentine sample was taken (equation (1)) based on the measurement from the enamel root junction. Next, we adjusted the position of the $\beta$ values by 6 months, assuming that the crown forms over 12 months (equation (2)). Finally, we calculated the relative position of the bioapatite sample, assuming the mineralization of the crown begins around 12 months (equation (3)). These temporal calculations are estimates, and have not been used in Fig. 5.

$$\% \text{ crown height} = \text{sample position/crown height} \times 100, \qquad (1)$$

$$(1 - \% \text{crown height}) \times 12 + 6, \qquad (2)$$

$$(1 - \% \text{crown height}) \times 12 + 12. \qquad (3)$$

Statistical analysis and graphic production were carried out using the free platform R program[77]. All maps and plots were produced using the R library ggplot2 (ref. [80]). River courses were sourced from Natural Earth's 'Rivers, Lake Centerlines 3.0.0' dataset[81]. The LBK outline was created by D.G.

### Reporting summary

Further information on research design is available in the Nature Portfolio Reporting Summary linked to this article.

## Data availability

Data are available in the Supplementary Information.

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

## Acknowledgements

We thank the European Research Council for funding an advanced grant (NeoMilk, FP7-IDEAS-ERC/324202, R.P.E.) financing a postdoctoral contract to R.E.G. and M.R.-S. and a PhD to I.P.K. and E. Casanova. M.R.-S. thanks the Royal Society for funding her Dorothy Hodgkin Fellowship (DHF\R1\180064 and RGF\EA\181067), V.B. thanks the OPRDE MEYS project (CZ.02.1.01/0.0/0.0/16_019/0000728) for funding her postdoctoral fellowship and A.M. thanks the Polish National Science Centre (2017/25/B/HS3/01242). NERC (Reference: CC010) are thanked for partial funding of the National Environmental Isotope Facility (NE/V003917/1). The authors wish to thank the NERC (NE/V003917/1), the European Research Council (FP7-IDEAS-ERC/340923, R.D. Pancost) and the University of Bristol for funding GC–MS and GC–IRMS capabilities used for this work. P. Bickle (University of York, UK) and J. Smyth (UCD, Ireland) are thanked for their work in securing access to archaeological material and collection of ceramic material from some of the sites presented in this study. C. Gonçalves and J. Cascalheira (ICArEHB, Portugal) are thanked for support with GIS information and E. Halliman, G. Devevey (ICArEHB, Portugal) and C. Makarewicz (CAU, Germany) are thanked for their comments on earlier versions of the manuscript. We acknowledge D. Altoft, B. Banecki, L. Benson, J. Dunne, C. Maule and C. Walton-Doyle (University of Bristol, UK) for experimental lipid analyses.

## Author contributions

R.E.G., I.P.K., M.R.-S., M.B., J.-D.V. and R.P.E designed the research. R.E.G., I.P.K., E. Casanova, V.B. and M.R.-S. performed research and analysed the stable isotope results. D.F. performed stable isotopic analysis on bioapatite. R.E.G. performed statistical analysis and figure design/creation. M.Z. generated interpolation maps and palaeo-environmental data. R.E.G., I.P.K., M.R.-S., M.Z., M.B. and R.P.E. wrote the paper. All the other authors provided access to zooarchaeological and pottery samples and site information. All the authors approved the final manuscript.

## Funding

## Competing interests

The authors state there are no competing interests.

## Additional information

**Extended data** is available for this paper at https://doi.org/10.1038/s41559-024-02553-y.

**Correspondence and requests for materials** should be addressed to Rosalind E. Gillis.

[1]ICArEHB, Faculdade de Ciências Humanas e Sociais, Universidade do Algarve, Faro, Portugal. [2]Institute for Prehistoric and Protohistoric Archaeology, Kiel University, Kiel, Germany. [3]Archéozoologie, Archéobotanique: Sociétés Pratiques et Environnement (UMR 7209), CNRS–Muséum National d'Histoire Naturelle, Paris, France. [4]Organic Geochemistry Unit, School of Chemistry, University of Bristol, Bristol, UK. [5]Independent researcher, Milan, Italy. [6]Institute of Archaeological Sciences, Eötvös Loránd University, Budapest, Hungary. [7]CNRS–ARCHIMEDE (UMR 7044), University of Strasbourg, Strasbourg, France. [8]School of Engineering and Applied Science, Princeton University, Princeton, USA. [9]Department of Dairy, Fat and Cosmetics, University of Chemistry and Technology Prague, Prague, Czech Republic. [10]CEA–CNRS–UVSQ Laboratoire de Sciences du Climat et de l'Environnement (UMR 8212), Université Paris-Saclay, Gif-sur-Yvette, France. [11]LVR–State Service for Archaeological Heritage, Bonn, Germany. [12]Herman Ottó Museum, Miskolc, Hungary. [13]Institute of Archeology, University of Gdańsk, Gdańsk, Poland. [14]István Dobó Castle Museum, Eger, Hungary. [15]Leibniz–Zentrum für Archäologie, Johannes Gutenberg University, Mainz, Germany. [16]National Institute for Preventive Archaeological Research (INRAP), Université Paris 1 Panthéon-Sorbonne, Paris, France. [17]Institute of Archaeology, HUN-REN Research Centre for the Humanities, Centre of Excellence of the Hungarian Academy of Sciences, Budapest, Hungary. [18]Trajectoires (UMR 8215), Université Paris 1 Panthéon-Sorbonne, Paris, France. [19]Hungarian National Museum, National Institute of Archaeology, Budapest, Hungary. [20]Department of Prehistoric and Historical Archaeology, University of Vienna, Vienna, Austria. [21]Independent researcher, Vienna, Austria. [22]Faculty of Archaeology, Adam Mickiewicz University, Poznań, Poland. [23]Comenius University, Archaeological Institute, Slovak Academy of Sciences, Nitra, Slovakia. [24]Department of Archaeology, University of Innsbruck, Innsbruck, Austria. [25]Institute of Archaeology, Maria Curie-Skłodowska University, Lublin, Poland. [26]Archaeological Heritage Office Saxony, Dresden, Germany. [27]Institute of Archaeology and Museology, Masaryk University, Brno, Czech Republic. [28]Faculty of Archaeology, Leiden University, Leiden, The Netherlands. [29]Present address: Department of Radiation Dosimetry, Institute of Nuclear Physics of the Czech Academy of Sciences, Prague, Czech Republic. [30]Present address: Department of History, Palacký University, Olomouc, Czech Republic. ✉e-mail: rosalind.gillis@dainst.de

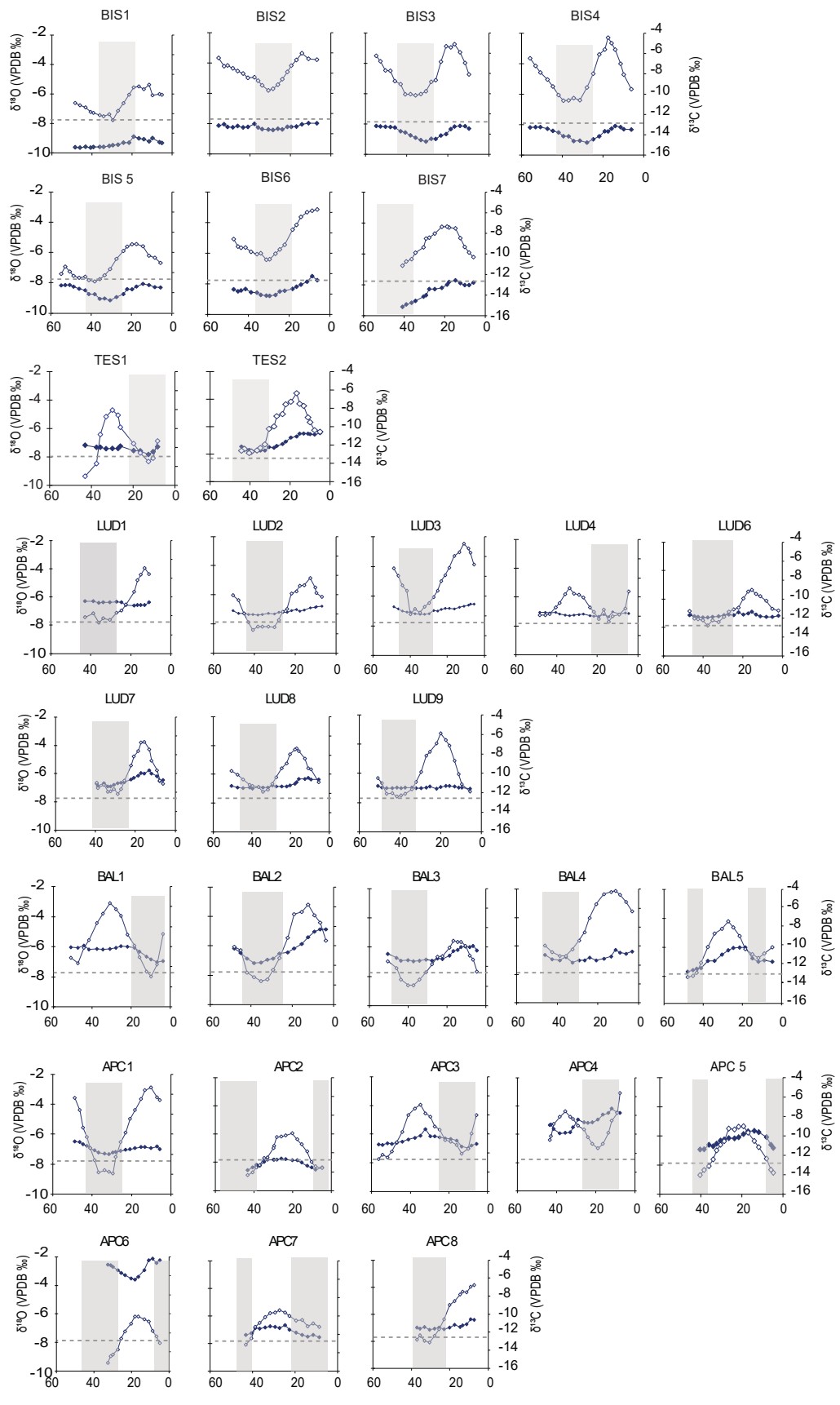

**Extended Data Fig. 1 | Bioapatite results.** The δ18O and δ13C values from sampled teeth from Bischoffscheim (BIS), Těšetice-Kyjovice (TES), Ludwinowo 7 (LUD), Apc-Berekalja (APC) and Balatonszárszó (BAL). The grey box represents winter/cold season based on low δ18O value, while the dotted line is the hypothetical forest signal.

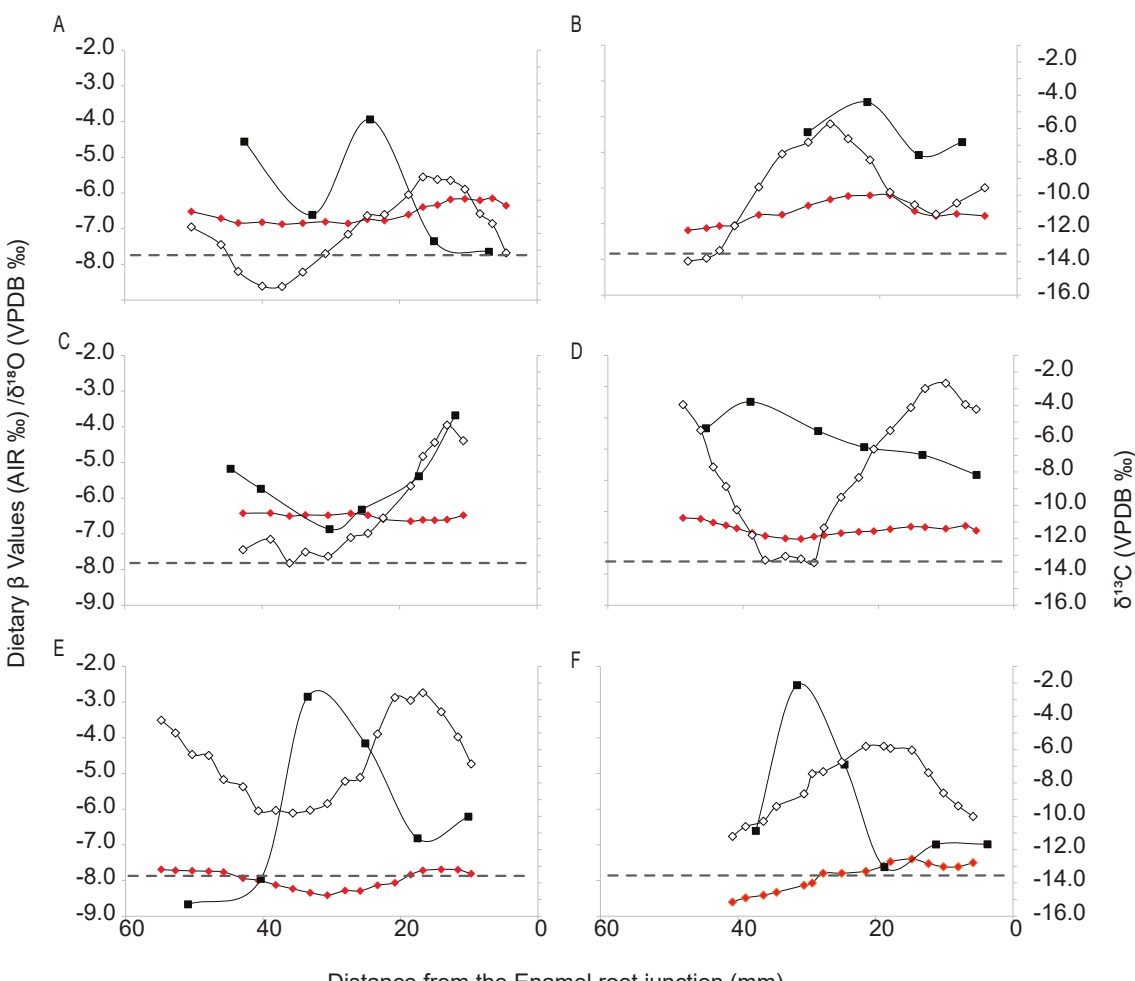

**Extended Data Fig. 2 | Uncorrected version of Fig. 5.** Raw values of bioapatite δ¹³Cbioap (red diamonds), δ¹⁸O (open diamonds) and dentine β values (black squares) from cattle molars: **A**. BAL3; **B**. BAL5; **C**. LUD1; **D**. APC1; **E**. BIS3; **F**. BIS4. The grey dotted line is the upper limit of both dietary β values for woody plants (−7.7‰ ref.45) and hypothetical δ¹³C value for forest dwelling herbivore (−13.2‰, based on −27.7‰ adjusted for Δbioap-diet by −14.5‰[59]).

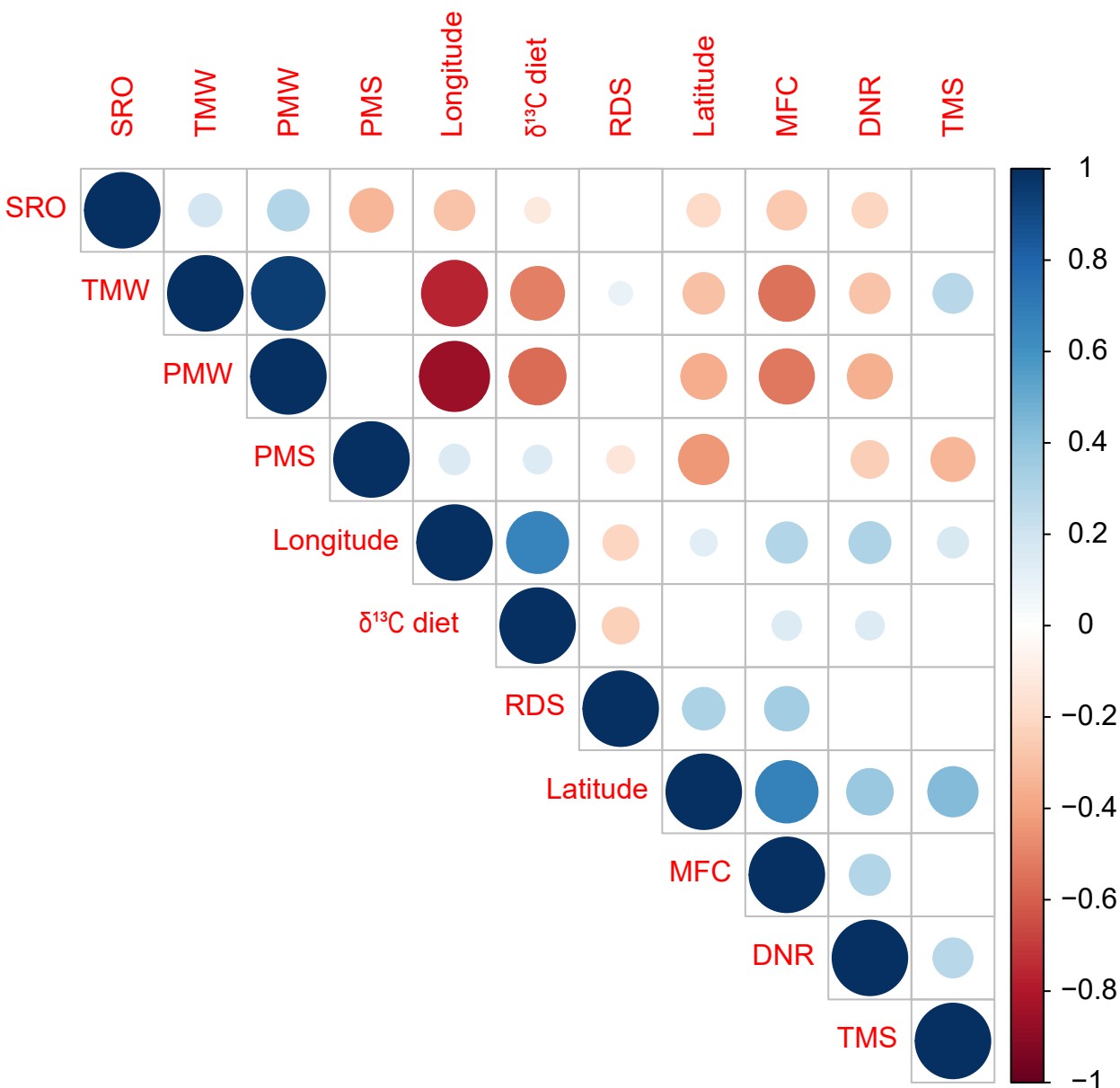

**Extended Data Fig. 3 | Correlation Plot.** The correlation result from the proxies: Longitude; Latitude; Strahler stream order (SSO); mean forest cover (MFC); mean precipitation in summer (PMS); and winter (PMW), mean temperature in summer (TMS) and winter (TMW); δ¹³C diet values; river drainage systems (RDS); distance to nearest river (DNR).

| | |
|---|---|

# Reporting Summary

## Statistics

For all statistical analyses, confirm that the following items are present in the figure legend, table legend, main text, or Methods section.

| n/a | Confirmed | |
|---|---|---|
| ☐ | ☒ | The exact sample size (*n*) for each experimental group/condition, given as a discrete number and unit of measurement |
| ☐ | ☒ | A statement on whether measurements were taken from distinct samples or whether the same sample was measured repeatedly |
| ☐ | ☒ | The statistical test(s) used AND whether they are one- or two-sided<br>*Only common tests should be described solely by name; describe more complex techniques in the Methods section.* |
| ☐ | ☒ | A description of all covariates tested |
| ☒ | ☐ | A description of any assumptions or corrections, such as tests of normality and adjustment for multiple comparisons |
| ☐ | ☒ | A full description of the statistical parameters including central tendency (e.g. means) or other basic estimates (e.g. regression coefficient) AND variation (e.g. standard deviation) or associated estimates of uncertainty (e.g. confidence intervals) |
| ☐ | ☒ | For null hypothesis testing, the test statistic (e.g. *F*, *t*, *r*) with confidence intervals, effect sizes, degrees of freedom and *P* value noted<br>*Give P values as exact values whenever suitable.* |
| ☒ | ☐ | For Bayesian analysis, information on the choice of priors and Markov chain Monte Carlo settings |
| ☒ | ☐ | For hierarchical and complex designs, identification of the appropriate level for tests and full reporting of outcomes |
| ☒ | ☐ | Estimates of effect sizes (e.g. Cohen's *d*, Pearson's *r*), indicating how they were calculated |

*Our web collection on statistics for biologists contains articles on many of the points above.*

## Software and code

Policy information about availability of computer code

| Data collection | Where relevant we have described how the data was collected. |
|---|---|
| Data analysis | Yes, full description of platforms and programs used to preform the analysis |

For manuscripts utilizing custom algorithms or software that are central to the research but not yet described in published literature, software must be made available to editors and reviewers. We strongly encourage code deposition in a community repository (e.g. GitHub). See the Nature Portfolio guidelines for submitting code & software for further information.

## Data

Policy information about availability of data

All manuscripts must include a data availability statement. This statement should provide the following information, where applicable:

- Accession codes, unique identifiers, or web links for publicly available datasets
- A description of any restrictions on data availability
- For clinical datasets or third party data, please ensure that the statement adheres to our policy

| |
|---|
| The data is avaiable as Supplementary tables |

March 2021

## Human research participants

Policy information about studies involving human research participants and Sex and Gender in Research.

| | |
|---|---|
| Reporting on sex and gender | N/A |
| Population characteristics | N/A |
| Recruitment | N/A |
| Ethics oversight | N/A |

Note that full information on the approval of the study protocol must also be provided in the manuscript.

# Field-specific reporting

Please select the one below that is the best fit for your research. If you are not sure, read the appropriate sections before making your selection.

☐ Life sciences  ☐ Behavioural & social sciences  ☒ Ecological, evolutionary & environmental sciences

For a reference copy of the document with all sections, see nature.com/documents/nr-reporting-summary-flat.pdf

# Ecological, evolutionary & environmental sciences study design

All studies must disclose on these points even when the disclosure is negative.

| | |
|---|---|
| Study description | The study used stable isotopic measurements of carbon, oxygen and nitrogen from ancient cattle bioapatite, dentine and collagen as well as ruminant dairy fats collected from prehistoric pottery. The relationship between climate and environment was explored using paleoenvironmental proxies. |
| Research sample | The research sample consisted of stable isotopic measurements of carbon, oxygen and nitrogen from ancient cattle (8000yrs old) bioapatite, dentine and collagen as well as ruminant dairy fats collected from prehistoric pottery (c.8000yrs old). |
| Sampling strategy | Samples were selected for good preservation of skeletal features. Pottery samples focused on non restored rim sherds. |
| Data collection | The data was collected by Melanie Roffet-Salque, Iain Kendall, Marco Zanon, Veronika Brychova, Arkadiusz Marciniak, Emmanuelle Casanova and Rosalind Gillis, during the laboratory and data analysis. |
| Timing and spatial scale | No repetition was made of stable isotopic measurements. These measurements are representable of a temporal period of between 5900BC and 5400BC cal. |
| Data exclusions | N/A |
| Reproducibility | The experiment design and methodology is well described and established. |
| Randomization | N/A |
| Blinding | N/A |

Did the study involve field work?  ☒ Yes  ☐ No

# Field work, collection and transport

| | |
|---|---|
| Field conditions | Archaeological material was collected during excavations that have taken place over the last 30 years. The material was then collected by Rosalind Gillis and Melanie Roffet-Salque from various depots and museums across central Europe. |
| Location | Central Europe |
| Access & import/export | All of the material has been exported with full permission of local authorities and excavators, most of whom are co-authors on the paper. All studied materials are currently being repatriated where possible. |
| Disturbance | N/A |

# Reporting for specific materials, systems and methods

We require information from authors about some types of materials, experimental systems and methods used in many studies. Here, indicate whether each material, system or method listed is relevant to your study. If you are not sure if a list item applies to your research, read the appropriate section before selecting a response.

## Materials & experimental systems

| n/a | Involved in the study |
|-----|----------------------|
| ☒ | Antibodies |
| ☒ | Eukaryotic cell lines |
| ☐ | ☒ Palaeontology and archaeology |
| ☐ | ☒ Animals and other organisms |
| ☒ | Clinical data |
| ☒ | Dual use research of concern |

## Methods

| n/a | Involved in the study |
|-----|----------------------|
| ☒ | ChIP-seq |
| ☒ | Flow cytometry |
| ☒ | MRI-based neuroimaging |

## Palaeontology and Archaeology

| | |
|---|---|
| Specimen provenance | All of the material has been exported with full permission of local authorities and excavators, most of whom are co-authors on the paper. All studied materials are currently being repatriated where possible. Given the large number of sites, it is not possible to provide full documentation and is available on request. |
| Specimen deposition | All studied materials are currently being repatriated where possible. Cattle teeth are well preserved and can be used for future analysis. All stable isotopic results are available with this publication. |
| Dating methods | Radiocarbon dates were provided by the excavators and are summarized in Supplementary table 1. |

☒ Tick this box to confirm that the raw and calibrated dates are available in the paper or in Supplementary Information.

| | |
|---|---|
| Ethics oversight | N/A |

Note that full information on the approval of the study protocol must also be provided in the manuscript.

## Animals and other research organisms

Policy information about studies involving animals; ARRIVE guidelines recommended for reporting animal research, and Sex and Gender in Research

| | |
|---|---|
| Laboratory animals | N/A |
| Wild animals | N/A |
| Reporting on sex | N/A |
| Field-collected samples | Ancient cattle and deer teeth and bone collected from archaeological sites. |
| Ethics oversight | N/A |

Note that full information on the approval of the study protocol must also be provided in the manuscript.

