## [Peer Review File · Nature Ecology & Evolution]

Peer Review Information

Journal: Nature Ecology & Evolution

Manuscript Title: Diverse Prehistoric Cattle Husbandry Strategies in the Forests of Central Europe

Corresponding author name(s): Rosalind E. Gillis

Editorial Notes:

Reviewer Comments & Decisions:

Decision Letter, initial version:

11th May 2022

Dear Dr Gillis,

First, I'd like to apologise for the delay in getting a decision to you on your manuscript entitled "Forest Ecosystems and Evolution of Cattle Husbandry Practices of the Earliest Central European Farming Societies"--one of the reviewers fell out of contact, but we feel we can proceed with a decision based on the two reviewer reports attached. The reviewers have raised a number of concerns which will need to be addressed before we can offer publication in Nature Ecology & Evolution. We will therefore need to see your responses to the criticisms raised and to some editorial concerns, along with a revised manuscript, before we can reach a final decision regarding publication.

The reviewers are primarily concerned with presentation and interpretation at this stage and suggest some restructuring (in addition to technical comments) to better hone the manuscript's findings and bring clarity.

We therefore invite you to revise your manuscript taking into account all reviewer and editor comments. Please highlight all changes in the manuscript text file.

* Include a "Response to reviewers" document detailing, point-by-point, how you addressed each reviewer comment. If no action was taken to address a point, you must provide a compelling argument.

2This response will be sent back to the reviewers along with the revised manuscript.

* If you have not done so already please begin to revise your manuscript so that it conforms to our Article format instructions at <http://www.nature.com/natecolevol/info/final-submission>. Refer also to any guidelines provided in this letter.

[REDACTED]

Nature Ecology & Evolution is committed to improving transparency in authorship. As part of our efforts in this direction, we are now requesting that all authors identified as 'corresponding author' on published papers create and link their Open Researcher and Contributor Identifier (ORCID) with their account on the Manuscript Tracking System (MTS), prior to acceptance. ORCID helps the scientific community achieve unambiguous attribution of all scholarly contributions. You can create and link your ORCID from the home page of the MTS by clicking on 'Modify my Springer Nature account'. For more information please visit www.springernature.com/orcid.

2[REDACTED]

Reviewer expertise:

Reviewer #1: Neolithic/Bronze Age archaeology

Reviewer #2: Stable isotope and residue analysis

Reviewers' comments:

Reviewer #1 (Remarks to the Author):

This is an interesting submission, which aims at systematically assess the relationship between cattle management during the LBK and forest cover across a large part of Europe. The paper is very dense and data-rich, so that some sections are difficult to read. While the discussion offers some key insights, there is a major discrepancy between the stated goals of the submission (systematic analysis of the data using a combination of statistical approaches at an explicit large scale), and the discussion which largely relies upon a case-by-case approach. In this sense, the paper requires further restructuring prior to publication.

The overall design of the research is well made, and relies upon a synthesis of various stable isotope measurements, and various environmental proxies. On several occasions, I found myself writing down questions regarding the processing of the latter data, to then find it in the methods. Though I understand this structure, it does not help comprehension of a very dense text, and clearer links between text and methods should be considered. From a technical point of view, a lot of the analysis, especially spatial, relies upon interpolated data, with use of IDW as a method. I would expect, only if briefly, a justification of this choice instead of more performing algorithms such as kriging. The methods section also notes that the interpolation for the stable isotopes values is made on the mean. Have the authors checked, for all instances, that the mean (rather than the median) is an appropriate measure. While I understand the use of a single value, it also erases a lot of the uncertainty and variability inherent to the original data, and I would like to know if the authors have considered ways to tackle this problem.

A lot of the analysis relies upon correlation analysis between stable isotope values and environmental proxies. Regardless of the results, I would have expected, as is standard is other techniques such as

3species distribution modelling, that the authors had first check the level of auto-correlation between the environmental values themselves. Was this done or not?

As indicated by the interpolated maps, there is a clear W-E gradient in the stable isotopes values, but, unless mistake of mine this is hardly a novel result and was highlighted by Hedges and colleagues over 10 years ago; the ref is present in the text, but this seems to minimise the originality of the present result.

As previously mentioned, the discussion starts by stressing the lack of overall clear systematic statistical correlation between environmental variables and varied stable isotopes measures. As such, this is really interesting and merits to be highlighted. However, this takes the bulk of the paper, and the rest of the discussion then proceeds in a rather old-fashioned case-by-case summary of the data to support the overall interpretation. There is thus a major imbalance at the core of the structure of the paper, which needs to be re-assessed. I am not doubting the general reading of the data, but simply pointing an inadequacy between methods and interpretations, especially so as the last part of the discussion is then methodologically very very traditional, and for instance lacking any strong statistical component. I would like the authors to consider this remark and re-assess accordingly the overall messaging and, possibly, presentation of the results which actually lead to the interpretation. Or, the other way around, what was actually gained by the heavy statistical analysis which forms the core of the paper?

Reviewer #2 (Remarks to the Author):

This study relates inferred cattle diet d13C values to various palaeoenvironmental and palaeoclimate proxies and uses d13C, d18O and dietary D15Nglx-Phe values of cattle teeth to infer the importance of forest pasturing/foddering. While this provides interesting insights into cattle husbandry practices in the LBK, I struggled a bit to identify the main “take home” messages from the study. I therefore suggest the following revisions to make the article suitable for publication.

The observation that bone collagen d13C values decrease from west to east in the LBK has already been made in Whittle & Bickle 2013 and it would be useful for the authors to explicitly state what this study adds – i.e. that the trend is true of cattle in particular, that it holds with a much larger dataset, that it is not related to MFC.

I also think the manuscript would benefit from more clarity on what the authors define as evidence for cattle being pastured in/having received fodder from the forest. This might be because there are different forms of data available for different sites, making it difficult to directly compare them. I wonder whether a figure that shows the inferred diet d13C values for cattle from all of the sites

mentioned in the Discussion would be useful, since some of these sites do not have bioapatite $\delta^{13}\text{C}$ and $\delta^{18}\text{O}$ or Beta values. I appreciate the attempt to use the inferred diet $\delta^{13}\text{C}$ range of deer and Beta values to define forest husbandry, but there is a case where an individual has relatively low bioapatite $\delta^{13}\text{C}$ values (APC2) but relatively high Beta values and cases where individuals have relatively high $\delta^{13}\text{C}$ values (BAL2 and BAL3) but relatively low Beta values and there is no attempt to explain this discrepancy.

Also, given the decrease in bone collagen $\delta^{13}\text{C}$ values from west to east in cattle and herbivores in general, have the authors considered that the $\delta^{13}\text{C}$ values of deer bone collagen could also change with longitude? Thus the use of a single interval of $\delta^{13}\text{C}$ values to define “forest feeding” across the whole LBK distribution may not be appropriate.

Finally, Figure 4 and the accompanying description of results focuses quite strongly on the relationship between amplitude of $\delta^{13}\text{C}$ and Beta values and longitude, but there is no subsequent discussion or attempted explanation of such trends.

In general, I think the discussion could be clearer and the main conclusions made more explicit. It is easy for the reader to get lost because of the range of data presented, but I do think that there are interesting conclusions to be drawn.

I also make more detailed comments/suggested modifications below:

Throughout the text, delta values need to be italicised.

L.61: replace “archaeology” with “archaeological”

L.62: insert “animal” before “husbandry” to make this clearer

L.79 and 80: “and” should replace the hyphen and “to” between dates

L.81: delete “time”

L.95: insert “a” before “diverse”

L.98: insert commas around “depending on their canopy structure and composition”

L.104: replace “ratios” with “values” since delta values are a ratio of a ratio

L.106: replace “relative negative” with “relatively low”. All of the d13C values in these cases are negative.

L.113: replace “is” with “are”. Are they related to or derived from consumed fodder?

L.118: You need to define d18O

L.119-120: Can you explain this in more detail? How can the d13C and d18O values be used to identify winter foddering?

L.124: You need to define d15N

L.142: should read “reconstructions”

Figure 2A: Could the number of d13C values that each material represents be included in the legend?

L.194: Define RMSE

L.202-204: Figure 3 doesn’t allow you to see that there is a decrease in the diet d13C values from east to west, since they are arranged in order of MFC rather than longitude.

L.213: What values? In Figure 4B, there looks to be one value that is greater than 3.0 per mil.

Figure 4: I think the boxes are labelled incorrectly – should 4B be 4C and vice versa? And I can’t see the grey boxes corresponding to deer bone collagen and woody plants, respectively.

L.220: “below the upper limit for diet based on woody plants”. This phrase is confusing. Perhaps it is because the grey box denoting the upper limit is missing from Figure 4, but I’m unsure as to how you calculate this and what it means in terms of foddering/herding strategy. Does this mean that for all other individuals, their Beta values fall outside the range of woody plants? If this is the case, I think it would be better to re-phrase it in these terms.

L.236: The subsequent interpretations in terms of the season of dietary inputs hinge on a delay of 6 months between isotopic signals of dentine and enamel, but the results are presented as distance from

enamel-root junction in mm. There therefore needs to be some additional information about the formation rate of each tissue to convert distance to months. Could months be included underneath distance on the x-axis? It would be even clearer if the dentine Beta values were adjusted on the x-axis to account for the different formation rate, but I acknowledge the authors' preference to display the unconverted data – perhaps include as a supplementary figure?

L.239: Instead of “indicating of woody plants”, rather “indicating the greatest potential contribution of woody plants to the diet”.

L.242: “coincide” rather than “coincided”

Figure 5. It is interesting to note that it is only in the cases of the individuals where consumption of woody plants is implied (5A, B, E) that the lowest Beta values coincide with the lowest d13C and d18O values (accounting for a difference in dentine and enamel formation rate). Perhaps this could be an additional source of evidence for seasonal consumption of woody fodder, given ethnographic evidence that it would have been preferentially given in the winter months?

L.272-274: Does the correlation observed in this study agree with the expected trend based on knowledge of 13C discrimination? Just referring to a “correlation” doesn't indicate whether there is a positive or negative relationship between diet d13C values and precipitation and temperature.

L.273: replace “pervious” with “previous”

L.275: What factors relating to palaeoclimate were investigated in this study?

L.279-293: But the positive correlation between longitude and diet d13C values is not only seen in the diet d13C values derived from pottery lipids. Therefore the trend seems to be consistent regardless of proxy. Perhaps more useful to note is that the trend is observed in all proxies, which suggests that the differences in diet are consistent throughout the year – during lactation as well as more generally. When is lactation in these cattle? Can some inference be made about the time of year that is reflected in the pottery lipid d13C values?

L.286: d13C values are not enriched – they are either higher or pig fat is 13C-enriched

L.286-288: This sentence is unclear. “pottery lipids” needs to be moved next to “faunal assemblages”. It would also be clearer if you explicitly state that the observed east-west trend in lipid d13C values cannot

be explained by the ubiquity of pig bones.

L.299: delete the comma after “BC”

L.306: specify that evidence for woody forage was found at Balatonszárszó

L.306-308: Do these differences exist between inferred d13C values of diet rather than between the two proxies? I would also argue whether the difference is real, given the large standard deviation in bioapatite d13C values. Also, you can identify seasonal differences in bioapatite d13C values, so can't you match up the d13C values of diet derived from dairy lipids and from bioapatite during the lactation period? These should be expected to match.

L.314: change to “relatively high diet d13C values compared to the reference for...”

L.315: change to “Compound specific and bulk collagen d13C values...”

L.318: insert “the” before “subsequent”

L.318-319: Specify what about the dairy lipids and collagen indicate this. Relatively low d13C values?

L.323-326: This seems to contradict the previous statement – how are you identifying browsing in the forest if not based on d13C values that are consistent with a forest dwelling herbivore? And you state that the individual at Chotěbudice likely received leafy hay fodder.

L.327-336: The interchangeable use of north and south and Haut-Rhin and Bas-Rhin is confusing here for those unfamiliar with the regions. I think better to stick with one of these.

L.328: What is the reference for forest type?

L.330: So is Bischoffsheim in the Bas-Rhin region?

L.333: What do the relatively high d13C values imply?

L.340: What do you mean by comparison with Bischoffsheim? You mean the similarity in bone collagen d13C values? Reiterate what was observed at Bischoffsheim.

L.342-343: Is comparing dairy and collagen d13C values meaningful? If they are similar does this imply a similar d13C value of diet? This needs to be specified here. And what are the implications of this observation? Does it suggest that cattle dairy fats were the main source of pottery lipids?

L.343-345: If the above is true, this sentence should start with “However”. This sentence should be re-phrased, perhaps: “However, the possibility that cattle were the main source of dairy lipids should be...”

L.347: “fattened” instead of “fatten”

L.348: insert “since” before “winter”

L.360: who are “them”?

L.364: Is there a word missing after “forest”?

L.369: replace “lead” with “led”

L.371-375: This study does not demonstrate that pasturing of animals in forests contributed to the opening-up of these environments. There are two separate lines of evidence here. This study demonstrates that some animals were pastured in forests and pollen evidence shows that forests opened up in this time period. You can suggest that the two are linked, but you need to nuance these sentences.

L.382-383: Define Glx and Phe. For which AA does the quoted trophic offset refer to? You should give the trophic offsets of both AAs. I’m not sure that reference 49 is correct here.

L.398-399: What is the justification for using the d13C value of the C16 fatty acid rather than that of the C18 fatty acid?

L.484: What were the accuracy and precision of the Glx and Phe d15N values of the standards?

L.521: Should read “Cheryl”

*****END*****

Author Rebuttal to Initial comments

Reviewers' comments:

Reviewer #1 (Remarks to the Author):

This is an interesting submission, which aims at systematically assess the relationship between cattle management during the LBK and forest cover across a large part of Europe.

The paper is very dense and data-rich, so that **some sections are difficult to read**. While the discussion offers some key insights, there is a major discrepancy between the stated goals of the submission (systematic analysis of the data using a combination of statistical approaches at an explicit large scale), and **the discussion which largely relies upon a case-by-case approach**.

We have revised the introduction to more clearly state our aims (investigate pasture and forage practices adopted by pioneer farming communities in central Europe and discuss the impact of these strategies on cattle herds, human societies and forest ecosystems), which clearly chimes with the discussion (impact of forest foddering on cattle herds, human societies and forest ecosystems). The case by case section has been removed and summarised to produce a more synthetic perspective of the results.

In this sense, the paper requires further restructuring prior to publication. The overall design of the research is well made, and relies upon a synthesis of various stable isotope measurements, and various environmental proxies. On several occasions, I found myself writing down questions regarding the processing of the latter data, to then find it in the methods. Though I understand this structure, it does not help comprehension of a very dense text, and clearer links between text and methods should be considered.

We have increased the number of 'sign posts' to help the reader understand how the data was processed. The supplementary data also includes a figure describing the theoretical background of the paper.

From a technical point of view, a lot of the analysis, especially spatial, relies upon interpolated data, with use of IDW as a method. I would expect, only if briefly, a justification of this choice

10instead of more performing algorithms such as kriging. The methods section also notes that the interpolation for the stable isotopes values is made on the mean. Have the authors checked, for all instances that the mean (rather than the median) is an appropriate measure. While I understand the use of a single value, it also erases a lot of the uncertainty and variability inherent to the original data, and I would like to know if the authors have considered ways to tackle this problem.

The interpolation method for stable isotopes used as visual aid we do not extract SI data from the interpolation maps in Figure 2. Concerning the extraction of site specific paleoclimate data from interpolation maps based on Sanchez-Goñi et al. 2016. The selection of the interpolation function relies also on the quality of the dataset. Kriging is a flexible interpolation algorithm that can indeed outperform the more simple IDW approach. Differences in performance depend nonetheless on the structure of the underlying dataset and should not be necessarily taken for granted.

To evaluate the differences between the two interpolation procedures, we implemented ordinary kriging through the R package gstat 2.0-6. The R package automap 1.0-16 was used to determine suitable variogram models for all climate variables.

Kriging-based paleoclimate variable were extracted for each site and compared with their IDW-based counterpart, showing a high degree of similarity between the two methodologies (see tab below).

Table 1 Pearson's correlation coefficient between kriging and IDW estimates for different palaeoclimatic variables

	Pearson's correlation coefficient	p-value
TMS	0.94	< 0.05
PMS	0.77	< 0.05
TMW	0.98	< 0.05
TMP	0.94	< 0.05

The correlation between kriging-based paleoclimate data and SI variables reveals few differences when compared with the correlation between IDW-based data and SI variables.

Table 2 comparison between the correlation coefficients of IDW- and kriging-based interpolations for different SI variables

		IDW	Kriging
lipids	TMS	0.21 (p<0.05)	-0.11 (p<0.05)
	PMS	0.18 (p<0.05)	0.46 (p<0.05)
	TMW	-0.63 (p<0.05)	-0.66 (p<0.05)
	PMW	-0.62 (p<0.05)	-0.63 (p<0.05)
collagen	TMS	0.03 (p>0.05)	-0.12 (p<0.05)
	PMS	0.20 (p<0.05)	0.44 (p<0.05)
	TMW	-0.37 (p<0.05)	-0.40 (p<0.05)
	PMW	-0.50 (p<0.05)	-0.54 (p<0.05)
bioapatite	TMS	0.14 (p>0.05)	-0.04 (p>0.05)
	PMS	-0.02 (p>0.05)	0.25 (p<0.05)
	TMW	-0.28 (p<0.05)	-0.32 (p<0.05)
	PMW	-0.36 (p<0.05)	-0.39 (p<0.05)

The performance of IDW and kriging are largely similar across most stable isotope sources and palaeoclimatic variables (tab.2), with the only meaningful difference represented by summer precipitation (PMS). Given the extensive similarities between the two interpolation models, we do not feel confident in preferring Kriging over IDW based on changes in PMS alone. Regardless,

12any difference between the two models does not affect the interpretation presented in the manuscript. Given these reasons, we prefer and IDW-based approach, which -given the available data- represents a simpler and effective solution.

A lot of the analysis relies upon correlation analysis between stable isotope values and environmental proxies. Regardless of the results, I would have expected, as is standard is other techniques such as species distribution modelling, that the authors had first check the level of auto-correlation between the environmental values themselves. Was this done or not?

We thank the reviewer for their comment and surprisingly auto-correction is rarely considered in archaeology. Spatial auto-correlation is unavoidable in ecology and when constructing models of explanation/prediction, it is important to check for auto-correlation of the residues. This leads to removal of data or proxies to reduce the auto-correlation. We cannot do this here as our dataset is constructed on a finite resource. Given this comment and those from the other reviewer, we have removed the PLSR as it does not bring any more clarity than the Pearson's regression. In our reporting of the correlation analysis, we discuss whether the correlation results are what we expect given our knowledge of discrimination of ^{13}C in different environments and climates. This reporting and the discussion focused on the choice of paleo-environmental proxies builds our argument that although climate appears to have an impact on $\delta^{13}\text{C}$ values, there remains a lot of variation within our datasets that cannot be explained by spatial or environmental variables alone.

As indicated by the interpolated maps, there is a clear W-E gradient in the stable isotopes values, but, unless mistake of mine this is hardly a novel result and was highlighted by Hedges and colleagues over 10 years ago; the ref is present in the text, but this seems to minimise the originality of the present result.

In response to this comment, we have summarised/discussed the previous studies that were carried out bone collagen (see introduction). Where our study is new is in the fact we extend the analysis to include values from lipids and bioapatite, and use CSIA-AA-d15N to verify leafy hay consumption as well as construct a detailed paleoenvironment dataset (MFC, paleoclimate) and modern rivers. The study of multiple tissues provides a holistic understanding of an organism's diet because different tissues reflect different time-scales. The resulting trend of the W-E depletion may be the same in both studies but our analysis goes beyond the initial Hedges

et al. study to explore the impact of forest foddering on herds, early farming communities and forested ecosystems.

As previously mentioned, the discussion starts by stressing the lack of systematic statistical correlation between environmental variables and various stable isotopes measures. As such, this is really interesting and merits to be highlighted. However, this takes the bulk of the paper, and the rest of the discussion then proceeds in a rather old-fashioned case-by-case summary of the data to support the overall interpretation. There is thus a major imbalance at the core of the structure of the paper, which needs to be re-assessed. I am not doubting the general reading of the data, but simply pointing an inadequacy between methods and interpretations, especially so as the last part of the discussion is then methodologically very very traditional, and for instance lacking any strong statistical component. I would like the authors to consider this remark and re-assess accordingly the overall messaging and, possibly, presentation of the results which actually lead to the interpretation. Or, the other way around, what was actually gained by the heavy statistical analysis which forms the core of the paper?

In response to the reviewers comment, we have removed the last case-by-case section of the discussion and summarised this section into the results (*integrated perspectives*). We have included a graph of all the sites discussed in this results section to provide more clarity. The comment about the lack of statistical component is confusing as it contradicts what the reviewer says about the text overall, in being statically heavy. However in light of the reviewers' comments, we have reduced the statically reporting by removing the PLSR and provided more interpretation of the results to support our findings and the discussion.

Reviewer #2 (Remarks to the Author):

This study relates inferred cattle diet $\delta^{13}\text{C}$ values to various palaeoenvironmental and paleoclimate proxies and uses $\delta^{13}\text{C}$, $\delta^{18}\text{O}$ and dietary $\text{D}_{15}\text{N}/\text{Glx-Phe}$ values of cattle teeth to infer the importance of forest pasturing/foddering. While this provides interesting insights into cattle husbandry practices in the LBK, **I struggled a bit to identify the main “take home” messages from the study.** I therefore suggest the following revisions to make the article suitable for publication.

The observation that bone collagen $\delta^{13}\text{C}$ values decrease from west to east in the LBK has already been made in Whittle & Bickle 2013 and it would be useful for the authors to explicitly

state what this study adds – i.e. that the trend is true of cattle in particular, that it holds with a much larger dataset, that it is not related to MFC.

In response to the reviewer 2, we have adjusted the introduction (Ln105-107) to make it clear that this had been observed in previous analysis as well as new findings that have not observed the same pattern in sheep (Ln107-108), establishing the point that a component of cattle diets included forest resources (Ln108-110). Our study brings new perspectives from other tissues to the initial research carried out by Hedges et al. 2013 with the lifeways project. Furthermore, we include Beta values, which are independent of the canopy effect and can provide direct evidence of forest fodder consumption. Finally we analyse this dataset within a paleoenvironmental dataset, unlike that of Hedges et al.

I also think the manuscript would benefit from more clarity on what the authors define as evidence for cattle being pastured in/having received fodder from the forest. This might be because there are different forms of data available for different sites, making it difficult to directly compare them.

We have included an extended figure (Extended figure 1) that defines the concept of the 'canopy effect' (EXFIG1A), herbivore signal (EXFIG1B), the enrichments between each tissue (EXFIG1C) and how we determine forest foddering in bioapatite sampling (EXFIG1D).

I wonder whether a figure that shows the inferred diet d13C values for cattle from all of the sites mentioned in the Discussion would be useful, since some of these sites do not have bioapatite d13C and d18O or Beta values.

The section of the discussion that the reviewer refers to has been condensed and moved to the results (*integrated perspectives*). We have included Fig 5 in this section as a focus for the final section of the results.

I appreciate the attempt to use the inferred diet d13C range of deer and Beta values to define forest husbandry, but there is a case where an individual has relatively low bioapatite d13C values (APC2) but relatively high Beta values and cases where individuals have relatively high d13C values (BAL2 and BAL3) but relatively low Beta values and there is no attempt to explain this discrepancy.

We thank the reviewer for this comments, we have addressed this omission. This point emphasises the importance of including Beta values in ruminant palaeodietary studies.

15Also, given the decrease in bone collagen d13C values from west to east in cattle and herbivores in general, **have the authors considered that the d13C values of deer bone collagen could also change with longitude?** Thus the use of a single interval of d13C values to define “forest feeding” across the whole LBK distribution may not be appropriate.

Deer values were not found to be significantly correlated with longitude, which is reported in the revised text in the first section of the results (Ln174). Furthermore, to address the reviewers concerns we have discussed the issues of using deer as a reference in the first section of the discussion (Ln327-333).

Finally, Figure 4 (now Fig.3) and the accompanying description of results focuses quite strongly on the relationship between amplitude of d13C and Beta values and longitude, but there is no subsequent discussion or attempted explanation of such trends.

We do not discuss longitude in this section associated with Fig 4 (now Fig 3). Perhaps the reviewer refers to our discussion with mean forest cover (MFC). We have included further observations of the bioapatite/dentine results.

In general, I think the discussion could be clearer and the main conclusions made more explicit. It is easy for the reader to get lost because of the range of data presented, but I do think that there are interesting conclusions to be drawn.

I also make more detailed comments/suggested modifications below:

These have been carried out. We thank the reviewer for taking the time to proof read the document. Concerning the italicisation of delta values, we have not done this as is not required by the journal nor is it an international convention.

L.118: You need to define d18O

This has been done see the introduction: *Oxygen isotopic ratios of local water sources are reflected in the $\delta^{18}O$ values³⁴. During winter, reduction in the evaporation transfer rate results in drinking water sources being depleted in ^{18}O , and the methods.*

L.119-120: Can you explain this in more detail? How can the d13C and d18O values be used to identify winter foddering?

This has been done and we have included Extended Fig. 1 that summarise the theoretical interpretation of forest canopy on bone, lipid and bioapatite. This is the first of its kind and we think one that the research community will be benefit from.

L.124: You need to define d15N

This is defined in the methods.

L.202-204: Figure 3 doesn't allow you to see that there is a decrease in the diet d13C values from east to west, since they are arranged in order of MFC rather than longitude.

This figure has been removed

L.213: What values? In Figure 4B, there looks to be one value that is greater than 3.0 per mil.

Figure 4: I think the boxes are labelled incorrectly – should 4B be 4C and vice versa? And I can't see the grey boxes corresponding to deer bone collagen and woody plants, respectively.

L.220: “below the upper limit for diet based on woody plants”. This phrase is confusing. Perhaps it is because the grey box denoting the upper limit is missing from Figure 4, but I'm unsure as to how you calculate this and what it means in terms of foddering/herding strategy. Does this mean that for all other individuals, their Beta values fall outside the range of woody plants? If this is the case, I think it would be better to re-phrase it in these terms.

Fig. 4 is now Fig. 3. We have clarified that the values we are concerned with are $\delta^{13}\text{C}$ and addressed the error highlighted by the reviewer. This omission of the grey box and mis-numbering of the figures has been addressed.

L.236: The subsequent interpretations in terms of the season of dietary inputs hinge on a delay of 6 months between isotopic signals of dentine and enamel, but the results are presented as distance from enamel-root junction in mm. There therefore needs to be some additional information about the formation rate of each tissue to convert distance to months. Could months be included underneath distance on the x-axis? It would be even clearer if the dentine Beta values were adjusted on the x-axis to account for the different formation rate, but I acknowledge the authors' preference to display the unconverted data – perhaps include as a supplementary figure?

The methodology is described in the method section, with the original figure 5 included as Extended Fig. 3 with the synthetic version as Fig. 4.

Figure 5. It is interesting to note that it is only in the cases of the individuals where consumption of woody plants is implied (5A, B, E) that the lowest Beta values coincide with the lowest $\delta^{13}\text{C}$ and $\delta^{18}\text{O}$ values (accounting for a difference in dentine and enamel formation rate). Perhaps this could be an additional source of evidence for seasonal consumption of woody fodder, given ethnographic evidence that it would have been preferentially given in the winter months?

Yes this is also our interpretation. See the section in the results concerning individual analysis (Fig. 3). Leafy hay is also discussed in the introduction and the extended figure 1 describes how we can identify leafy hay in bioapatite results.

L.272-274: Does the correlation observed in this study agree with the expected trend based on knowledge of ^{13}C discrimination? Just referring to a “correlation” doesn’t indicate whether there is a positive or negative relationship between diet $\delta^{13}\text{C}$ values and precipitation and temperature.

We have revised this in the results section and clarified what we would expect for each of the paleo-environmental variables that we examine.

L.275: What factors relating to palaeoclimate were investigated in this study?

It is not clear whether the reviewer is referring to our study or that of Hedges et al. We state clearly that Hedges et al. 2013 used modern climatic values (Ln 310). If the reviewer's comment is concerned with our study, we describe all the paleoclimate variables in Ln137-141.

L.279-293: But the positive correlation between longitude and diet $\delta^{13}\text{C}$ values is not only seen in the diet $\delta^{13}\text{C}$ values derived from pottery lipids. Therefore the trend seems to be consistent regardless of proxy. Perhaps more useful to note is that the trend is observed in all proxies, which suggests that the differences in diet are consistent throughout the year – during lactation as well as more generally. When is lactation in these cattle? Can some inference be made about the time of year that is reflected in the pottery lipid $\delta^{13}\text{C}$ values?

We have deleted the section explaining the potential biases to the pottery lipids, and, as the reviewers proposed, focused on the point that the trend is observed in all proxies (Ln 338-339).

We thank the reviewer for the suggestion about lactation timings. It is difficult to make assumptions about the season of lactation from the pottery lipids without the use of deuterium isotopes. However, we note that if lactations were overlapping and occurring across two seasons then this may explain large variation in d13C values observed in pottery lipids (Ln356-358).

L.286-288: This sentence is unclear. “pottery lipids” needs to be moved next to “faunal assemblages”. It would also be clearer if you explicitly state that the observed east-west trend in lipid d13C values cannot be explained by the ubiquity of pig bones.

We have deleted this discussion about the lipids results since it is clear that lipid results are following the trends seen in collagen and bioapatite.

L.306-308: Do these differences exist between inferred d13C values of diet rather than between the two proxies? I would also argue whether the difference is real, given the large standard deviation in bioapatite d13C values. Also, you can identify seasonal differences in bioapatite d13C values, so can't you match up the d13C values of diet derived from dairy lipids and from bioapatite during the lactation period? These should be expected to match.

At the present moment, we cannot match up or make inferences between the bioapatite those from the lipids. However, we have noted that in areas where pottery lipids have a large variation that this may be the reflection of long/overlapping lactations.

L.315: change to “Compound specific and bulk collagen d13C values...”

This section has been removed.

L.318-319: Specify what about the dairy lipids and collagen indicate this. Relatively low d13C values?

This section has been deleted.

L.323-326: This seems to contradict the previous statement – how are you identifying browsing in the forest if not based on d13C values that are consistent with a forest dwelling herbivore? And you state that the individual at Chotěbudice likely received leafy hay fodder.

This section has been deleted.

L.327-336: The interchangeable use of north and south and Haut-Rhin and Bas-Rhin is confusing here for those unfamiliar with the regions. I think better to stick with one of these.

This section has been deleted.

L.333: What do the relatively high d13C values imply?

This section has been deleted.

L.340: What do you mean by comparison with Bischoffsheim? You mean the similarity in bone collagen d13C values? Reiterate what was observed at Bischoffsheim.

This section has been deleted.

L.342-343: Is comparing dairy and collagen d13C values meaningful? If they are similar does this imply a similar d13C value of diet? This needs to be specified here. And what are the implications of this observation? Does it suggest that cattle dairy fats were the main source of pottery lipids?

This section has been deleted.

L.343-345: If the above is true, this sentence should start with “However”. This sentence should be re-phrased, perhaps: “However, the possibility that cattle were the main source of dairy lipids should be...”

This section has been deleted.

L.371-375: This study does not demonstrate that pasturing of animals in forests contributed to the opening-up of these environments. There are two separate lines of evidence here. This study demonstrates that some animals were pastured in forests and pollen evidence shows that forests opened up in this time period. You can suggest that the two are linked, but you need to nuance these sentences.

We have revised this paragraph (now Ln 384-393) by describing the background to our perspective and highlighted that the non-uniformity of forest use as pasture may indicate that the modification process was not uniform across central Europe.

L.382-383: Define Glx and Phe. For which AA does the quoted trophic offset refer to? You should give the trophic offsets of both AAs. I'm not sure that reference 49 is correct here.

We have now defined $\Delta^{15}\text{N}_{\text{Glx-Phe}}$. The quoted offset refers to the $\Delta^{15}\text{N}_{\text{Glx-Phe}}$ value as a whole, which is the basis of the dietary β value. The specific individual AA offsets are not relevant here, but can be found in the now corrected reference.

L.398-399: What is the justification for using the d13C value of the C16 fatty acid rather than that of the C18 fatty acid?

We have added this in the main text "The C_{16:0} fatty acid was chosen over the C_{18:0} fatty acid in this study, as $\delta^{13}\text{C}_{16:0}$ values are reflecting the ruminant's diet, while $\delta^{13}\text{C}_{18:0}$ values are also reflecting tissue type (cite Copley et al. 2003, PNAS)."

L.484: What were the accuracy and precision of the Glx and Phe d15N values of the standards?

This has been added at the end of the section.

Decision Letter, first revision:

8th August 2022

Dear Dr Gillis,

Your manuscript entitled "Forest Ecosystems and Evolution of Cattle Husbandry Practices of the Earliest Central European Farming Societies" has now been seen by the same two reviewers, whose comments are attached. The reviewers feel that the manuscript has for the most part improved, but still have concerns that must be addressed before we can offer publication in Nature Ecology & Evolution.

We will therefore need to see your responses to the criticisms raised and to some editorial concerns, along with a revised manuscript, before we can reach a final decision regarding publication.

One issue mentioned by reviewer 1 in particular is that some of the results are disjointed from the

21introduction and discussion and could do with more thorough integration, and more significantly, that the interpretation of the results does not advance upon what was previously demonstrated for the field.

We therefore invite you to revise your manuscript taking into account all reviewer and editor comments. Please highlight all changes in the manuscript text file [OPTIONAL: in Microsoft Word format].

* If you have not done so already please begin to revise your manuscript so that it conforms to our Article format instructions at <http://www.nature.com/natecolevol/info/final-submission>. Refer also to any guidelines provided in this letter.

[REDACTED]

Nature Ecology & Evolution is committed to improving transparency in authorship. As part of our efforts in this direction, we are now requesting that all authors identified as 'corresponding author' on published papers create and link their Open Researcher and Contributor Identifier (ORCID) with their account on the Manuscript Tracking System (MTS), prior to acceptance. ORCID helps the scientific community achieve unambiguous attribution of all scholarly contributions. You can create and link your ORCID from the home page of the MTS by clicking on 'Modify my Springer Nature account'. For more information please visit www.springernature.com/orcid.

[REDACTED]

Reviewer expertise:

as before

Reviewers' comments:

Reviewer #1 (Remarks to the Author):

First of all, I would like to thank the authors for the efforts they have put in considering the majority of the comments made by both reviewers, and thus modifying accordingly their manuscript.

This being said, the manuscript still presents a few problems, and still cannot be accepted for publication. Detailed comments can be found hereafter:

- the authors seem not to have fully understood my comments about correlation between environmental factors. I was not referring to spatial auto-correlation (which is in itself an interesting topic), but to what extent the variables are correlated together, ie to what extent identifying similar r values for different proxies is meaningful at all. Looking for instance at the close r values for TMW and PMW, it is obvious that this is the same signal (as expected from a climatic point of view, and especially

23with simulated climatic values). Once more judging from the correlation values only, the same may hold for PMS and TMS, although it seems less straightforward;

- on line 188, the authors state that correlation between isotope values and MFC "perform well", pointing out the p-values. Rather than focusing on the latter, they should consider more closely their correlation values as, for instance a r value of -0.06 for bioapatite just highlights the lack of correlation (in a significant way as pointed out by the p-value!). Likewise, a r value of -0.29 is hardly that strong a signal. I also do not understand the comment by the authors stating that the lack of expected signal for dairy lipids may be related to the larger spatial distribution and size of the sample. Surely the fact that the expected signal is NOT found in the bigger dataset should be taken into consideration when assessing the robustness of the tested hypothesis...

- generally speaking, there is still only limited assessment of data structure and biases (this issue for stable isotopes being only admitted in line 328, deep into the discussion...). Likewise, there is no assessment of the range of values for each proxy being analysed, so that it is difficult to identify how robust each individual signal is, or is not.

Lastly, the text still requires editing, as I observed several typos (e.g. lines 75, 107, 359)

- section on integrated perspectives. The statement on line 267 is simply not supported by your results. The rest of the section is not "integrated", but simply a few, admittedly interesting, examples
- the discussion still feels poorly connected to the rest of the paper, in the sense that it makes very limited use of the - limited as well...- results. This is particularly evident in the concluding paragraphs starting at line 357

Reviewer #2 (Remarks to the Author):

I'm generally happy with how the manuscript has been revised, but think that it could benefit from a few more changes:

L.104: Needs a reference.

L.120-124: I would still appreciate a bit more clarity on this point. "consumption of winter foddering with summer collected leafy hay" – does this mean that cattle are foddered in the winter with summer collected leafy hay from forests? There is also no discussion of the other scenario in extended Figure 1 – whereby year-round grazing/browsing in forest pasture would result in lower $\delta^{13}\text{C}$ values during the summer.

L.194: The correlation between MFC and d13C values of bioapatite is barely negative (-0.06). In fact, all three proxies differ in the direction of correlation, which seems strange. This is discussed a little but then largely glossed over in the abstract and conclusion, which focus on the correlation between MFC and d13C values. I think a bit more transparency regarding this would be useful.

L.197-201: The correlations between d13C values and climatic proxies are much more consistent than the correlation with MFC, yet this is what you pull out as the pertinent factor. I would question this interpretation because of the differences in direction of correlation between MFC and d13C values.

Figure 4 caption - what does the dotted red line represent? There are no black filled squares in the figure that I can see.

L.255-256: References are needed.

In general, the manuscript would also benefit from another grammatical check as there are a number of typos/errors that make understanding the text difficult at times.

*****END*****

Author Rebuttal, first revision:

Dr. Rosalind Gillis
Abteilung Z / Referat
Naturwissenschaften
Im Dol 2-6
D 14195 Berlin

Tel.: +49 (0) 30 187711-308
rosalind.gillis@dainst.de

25Dear Reviewers,

We thank you for the time that you have given to reviewing our paper. We have made considerable changes in light of these comments and corrections in both rounds of reviews. These changes have considerably improved the paper by clarifying our methodology and the reporting of the results, which have made our conclusions clearer and stronger. In summary our conclusions, are that forests were used by the first farmers of central Europe for cattle grazing and fodder resources, although this practice varied across the region, with some areas being used more intensively than others. The use of forests for fodder and grazing would have a positive impact on cattle and human health. Our discussion has focused on this as we believe that our findings provide new perspectives on the evolution of prehistoric herds as well as the impact of this strategy may have had on forests ecosystems.

We would like to draw the reviewers' attention to the size of the dataset and spatial distribution of sampled sites, which was negatively commented on. The spatial distribution of any archaeological study of this nature is constrained by the preservation conditions. Archaeological remains are finite and the recovery and scientific examination of these materials depends on a long-chain of operators and funding. The NeoMilk project was privileged to have been well-funded thanks to the ERC Advance grant awarded to Prof Evershed. However, the bones and pottery that were studied during the NeoMilk project were provided via a network of goodwill cultivated by us and the distribution of sites was largely dependent on individual research and infra-structure programs. The *Linearbankeramik* sites are well-known for preserving very little organic remains, especially in the heart-land of the culture i.e. southern Germany. At present, this is the largest study of its kind for any archaeological culture.

Please find below our responses to individual reviewer comments:

Reviewer #1 (Remarks to the Author):

First of all, I would like to thank the authors for the efforts they have put in considering the majority of the comments made by both reviewers, and thus modifying accordingly their manuscript. This being said, the manuscript still presents a few problems, and still cannot be accepted for publication.

Detailed comments can be found hereafter:

- the authors seem not to have fully understood my comments about correlation between environmental factors. I was not referring to spatial auto-correlation (which is in itself an interesting

26topic), but to what extent the variables are correlated together, ie to what extent identifying similar r values for different proxies is meaningful at all. Looking for instance at the close r values for TMW and PMW, it is obvious that this is the same signal (as expected from a climatic point of view, and especially with simulated climatic values). Once more judging from the correlation values only, the same may hold for PMS and TMS, although it seems less straightforward;

We thank the reviewer for clarifying this point. Following on from this comment, we have integrated the results from the correlation between paleoenvironmental proxies within this section (ln: 203-235). The conclusion that emerges from these results is that there is strong correlation between winter climatic proxies while summer climatic proxies the correlation between each other and that of longitude and forest cover is weak to non-existent.

- on line 188, the authors state that correlation between isotope values and MFC "perform well", pointing out the p-values. Rather than focusing on the latter, they should consider more closely their correlation values as, for instance a r value of -0.06 for bioapatite just highlights the lack of correlation (in a significant way as pointed out by the p-value!). Likewise, a r value of -0.29 is hardly that strong a signal. I also do not understand the comment by the authors stating that the lack of expected signal for dairy lipids may be related to the larger spatial distribution and size of the sample. Surely the fact that the expected signal is NOT found in the bigger dataset should be taken into consideration when assessing the robustness of the tested hypothesis...

The reviewer is correct that -0.29 is not a strong result. There are multiple factors (climatic etc.) that may have had an influence on stable carbon isotope results. Therefore this result is an indication of the proportion of influence that forest cover may have potentially had on stable isotope results. The positive correlation between MFC and dairy lipids suggests a different relationship between MFC and $\delta^{13}\text{C}$. The revision of the results in light of the reviewers' comments, have highlighted this result. The lack of a negative correlation for dairy lipids forms the foundation that in some areas where there was open pasture available, forests were deliberately and intensively used for pasture by farmers.

- generally speaking, there is still only limited assessment of data structure and biases (this issue for stable isotopes being only admitted in line 328, deep into the discussion...). Likewise, there is no assessment of the range of values for each proxy being analysed, so that it is difficult to identify how robust each individual signal is, or is not.

We thank the reviewer for this observation. In light of this, we have at the beginning of the results section provided a summary of the raw data as well as an assessment of the distribution of the sites and size of the datasets.

'The individual $\delta^{13}\text{C}$ values from cattle bone collagen, dental bioapatite and ruminant dairy lipids were: $-20.9 \pm 1.27\text{‰}$ (Nsites=24, N=292), $-11.5 \pm 1.53\text{‰}$ (based on mid-point, Nsites=7, Nteeth=46) and $-27.7 \pm 1.5\text{‰}$ (Nsites=45, N=352) respectively. The Upper Rhine and Upper/Middle Danube, Vistula and Oder drainage areas represented by four sites or more (Supplementary Table 1).'

And discuss the level of preservation of bone samples across the region and bias in the sampling:

"The poor preservation of bone collagen is evident in some regions, with no samples collected for bone collagen from sites in the Meuse, Middle Rhine and Vistula regions. The ruminant dairy lipid dataset is the largest and has the best regional distribution although there appears to be a bias towards the Oder region (Nsites=13; N=61) while a single site sampled in the Seine region, yielded 49 isotopic measurements. Dental remains often survive well but their recovery is subject to depositional and excavation practices. This restricts the availability of samples, as reflected in the small number of sites sampled for teeth. Overall, most of the regions discussed here have at least a single sampled site."

Lastly, the text still requires editing, as I observed several typos (e.g. lines 75, 107, 359) - section on integrated perspectives. The statement on line 267 is simply not supported by your results. The rest of the section is not "integrated", but simply a few, admittedly interesting, examples

We thank the reviewer for identifying the faults in the English language and we have revised and edited the text. While it is unclear what the reviewer is referring to in Ln267, we have removed the 'integrated perspectives' section as they do not add anything to our conclusions and reduces our over word count. Parts of this section have been integrated into the discussion.

- the discussion still feels poorly connected to the rest of the paper, in the sense that it makes very limited use of the - limited as well...- results. This is particularly evident in the concluding paragraphs starting at line 357

It is disappointing that the reviewer find our results limited. This is the most comprehensive dataset of its type for a single culture. The study was the result of a 5 year study and while we wish to have more data to analysis there is always a deadline and a budget. Again, it is not clear what the reviewer refers to with Ln357, which in past versions discussed the variation in forest types and how this may have impacted the results.

However, with the reviewers' comments in mind, we have revised the discussion to discuss the issues with sample distribution, issues with proxies and the datasets. We believe the discussion tackles issues with the proxies and data, and presents the problems with large-scale analysis of this kind. Our

discussion also discusses the positive and negative effects that forest grazing and fodder consuming in light of the results, namely LBK cattle in some areas were grazing and consuming forest fodder. Considering the scope of the journal, namely evolution and ecology, this discussion brings new perspectives to prehistoric animal husbandry that hitherto had not been considered and the potential impact of forest foddering on cattle herds, human societies and landscapes.

Reviewer #2 (Remarks to the Author):

I'm generally happy with how the manuscript has been revised, but I think that it could benefit from a few more changes:

L.104: Needs a reference.

We have adjusted this statement to be more specific '*Plant $\delta^{13}C$ values are directly influenced by photosynthesis, and local growing environments, particularly water availability^{32,33}.*' and added the reference.

L.120-124: I would still appreciate a bit more clarity on this point. "consumption of winter foddering with summer collected leafy hay" – does this mean that cattle are foddered in the winter with summer collected leafy hay from forests?

We believe that the extended Figure 1D, provides the clarification that the reviewer needs. Moreover, we have added, '*The density of deciduous forest canopies changes seasonally, reaching a maximum during late summer³⁷, and with plant foliar $\delta^{13}C$ values being at their lowest (Ln128-129)*'. This is when the most leaves are available and we believe it would have been integrated into their harvesting schedules as well.

There is also no discussion of the other scenario in extended Figure 1 – whereby year-round grazing/browsing in forest pasture would result in lower $d_{13}C$ values during the summer.

There is no evidence for it and this has been added to the text (Ln240; Ln 367/368).

L.194: The correlation between MFC and $d_{13}C$ values of bioapatite is barely negative (-0.06). In fact, all three proxies differ in the direction of correlation, which seems strange. This is discussed a little but then largely glossed over in the abstract and conclusion, which focus on the correlation between MFC and $d_{13}C$ values. I think a bit more transparency regarding this would be useful.

29We thank the reviewer for this comment. In light of this and the reviewer 1 comments, we have revised the results section concerning paleoenvironmental proxies. The revision now highlights this variation between datasets and MFC. We believe that this more transparent and clarifies our results. Furthermore, it emphasizes one of our conclusions that in some areas where there is open pasture available there is an intensive use of forests for cattle.

L.197-201: The correlations between d13C values and climatic proxies are much more consistent than the correlation with MFC, yet this is what you pull out as the pertinent factor. I would question this interpretation because of the differences in direction of correlation between MFC and d13C values.

Please see our comments to R1, moreover, as we state above we have revised this section and believe that it clear why we focus on the winter climate and MFC proxies. When consider together, the MFC proxy is the second most important variable in explaining our results in terms of order the most significance. It should be emphasised that this is the first paper of its type that considers the paleo-vegetation and has statistically assessed the impact of this proxy on stable isotope values.

Figure 4 caption - what does the dotted red line represent? There are no black filled squares in the figure that I can see.

We apologise for this unclear caption for the figure and it has been revised. The red dotted line is the curve based on dentine results.

L.255-256: References are needed.

This is not relevant now. But the statements concerning the timescales involved in different stable isotope signatures can be found in Ln 117-124.

'Milk C_{16:0} fatty acids largely reflect dietary isotopic signatures from the last days/weeks³⁴, whereas those of bioapatite and dentine are at an approximately monthly scale during the tooth development³⁵. Bone collagen stable isotope values average the dietary signature over the last months/years of an animal's life due to bone turnover³⁴. Compound-specific stable isotopic analysis (CSIA) of fatty acids (FA) from milk lipids trapped in pottery is a rich source of information about ruminant diets due to milk FAs directly being biosynthesised from dietary carbohydrates and FA³⁶.'

In general, the manuscript would also benefit from another grammatical check as there are a number of typos/errors that make understanding the text difficult at times.

We again apologise for these errors, and have taken considerable effort to revise and edit the text.

30Decision Letter, second revision:

19th December 2022

Dear Dr Gillis,

Your manuscript entitled "Forest Ecosystems and Evolution of Cattle Husbandry Practices of the Earliest Central European Farming Societies" has now been seen by the same two reviewers, whose comments are attached. The reviewers have raised a number of concerns which will need to be addressed before we can offer publication in Nature Ecology & Evolution. We will therefore need to see your responses to the criticisms raised and to some editorial concerns, along with a revised manuscript, before we can reach a final decision regarding publication.

As we discussed, please take some time to restructure the manuscript to bring out the key findings and more clearly explain the methodological approach.

We therefore invite you to revise your manuscript taking into account all reviewer and editor comments. Please highlight all changes in the manuscript text file.

* Include a "Response to reviewers" document detailing, point-by-point, how you addressed each

31reviewer comment. If no action was taken to address a point, you must provide a compelling argument. This response will be sent back to the reviewers along with the revised manuscript.

* If you have not done so already please begin to revise your manuscript so that it conforms to our Article format instructions at <http://www.nature.com/natecolevol/info/final-submission>. Refer also to any guidelines provided in this letter.

[REDACTED]

We hope to receive your revised manuscript within four to eight weeks. Given the time of year, I suggest adding at least two weeks onto that time schedule to take into account the end of year holidays. If you need more time beyond that, please let us know. We will be happy to consider your revision so long as nothing similar has been accepted for publication at Nature Ecology & Evolution or published elsewhere.

Nature Ecology & Evolution is committed to improving transparency in authorship. As part of our efforts in this direction, we are now requesting that all authors identified as 'corresponding author' on published papers create and link their Open Researcher and Contributor Identifier (ORCID) with their account on the Manuscript Tracking System (MTS), prior to acceptance. ORCID helps the scientific community achieve unambiguous attribution of all scholarly contributions. You can create and link your ORCID from the home page of the MTS by clicking on 'Modify my Springer Nature account'. For more information please visit www.springernature.com/orcid.

We look forward to seeing the revised manuscript and thank you for the opportunity to review your

32work.

[REDACTED]

Reviewer expertise:

as before

Reviewers' comments:

Reviewer #1 (Remarks to the Author):

First of all, I'd like to thank the authors for their patience and having made extensive efforts to integrate the previous comments by the reviewers.

As a result, I now consider that this paper is ready for publication

Reviewer #2 (Remarks to the Author):

My overarching concern (which seems to be more evident in this latest revision) is that the authors seem to contradict themselves throughout the manuscript.

In the Introduction, they focus on the importance of forest resources for cattle husbandry, but then show that there is a positive relationship between dairy lipid d13C values and MFC, which would seem to contradict an association between availability of forest resources and their consumption by cattle. They do not account for this relationship convincingly.

They maintain that low d13C values are due to consumption of woodland fodder, but then show using d15N-AA values that low d13C values are not consistently associated with consumption of lignin-rich plants.

The other assertions that they make in their abstract and conclusion – that “The long-term impact of forest pasture and fodder would have improved female cattle health, cattle breeding and milk availability for human consumption, as well as contributing to the emergence of man-modified forests.”

33derive from the results of other studies and not from this one.

I think the authors need to clarify the research questions that they seek to address with these new data and be consistent in the conclusions that they reach. I think they are trying to conflate too many factors: climate, MFC and foddering strategies, which is problematic because animal diet (as they note) is not a direct reflection of natural conditions but rather influenced by human management. I don't think they have yet succeeded in disentangling these factors.

There are still a considerable number of grammatical errors.

More detailed comments are below:

L.69-70: "Our analysis reveals that farmers practiced a diverse set of pasturing strategies with the intensive use of forested ecosystems in some areas for both graze and seasonal forage reflecting the adaptation of herding to new environments": Why does this indicate the adaptation to new environments? Wouldn't it rather reflect a particular husbandry practice being practiced regardless of environment?

L.79-80: This sentence doesn't make grammatical sense.

L.100: "waterlogged remains": Of what? Cattle dung? Was it also sheep and goat dung?

L.109: "photosynthesis": What do you mean by this? The photosynthetic pathway?

L.120: "biosynthesis route for individual tissues": ? Do you mean the different biosynthetic routing of nutrients into different tissues?

L.126: "reflect are at an approximate...": This doesn't make grammatical sense and also will surely depend on the sampling density.

L.139: "both sheep and cattle": How many individuals?

L.143-144: This sentence is very vague and it isn't clear how this is the case.

L.150: Are MFC predictions based on palynological data?

L.195: The distribution isn't global. Rather overall range?

L.236: “If forest resources were used for cattle forage”: Surely, it is actually if the relative importance of forest resources for cattle forage was related to their availability? I.e. if there was more forest, cattle would have eaten more plants influenced by the canopy effect?

L.238: $r=-0.06$: This isn't a correlation. It is basically showing that there is no relationship between MFC and $\delta^{13}C$ values.

L.245: “indication of cattle feeding in open water rich growing environments”: Why? What is the mechanism and its effect on plant $\delta^{13}C$ values?

L.246-247: “Overall, the correlation analysis with the paleoenvironment proxies suggest that winter climatic conditions followed by MFC had the greatest influence on plant foliar $\delta^{13}C$ values”: This assertion is dubious given the differing correlations between the different proxies. You need to try and explain the basis of this more convincingly.

L.250 onwards: This section provides much more convincing evidence for use of woody fodder and therefore use of forest resources. But it is based on a(n understandably) limited set of samples.

L.260: “potentially indicating different fodder types”: Or just access to plants growing in different levels of light intensity/water availability.

L.287: instead of “within cattle fodder”, consumed by cattle. My understanding is that fodder is food deliberately collected for and given to livestock, rather than everything consumed by them, which could include grazed vegetation.

L.296-298: Replace with: Figure 4A-F: Corrected combined stable isotopic results of incremental analysis of tooth enamel $\delta^{13}C$ values (grey diamonds, extended dashed grey line) and $\delta^{18}O$ values (white diamonds) and dentine AA- $\delta^{15}N$ values (red dotted line) from cattle teeth samples:

Figure 4A-F: How do you model the red dotted line? Shouldn't you show the determined values rather than extrapolate? Or at least show both.

L.318: “between diet $\delta^{13}C$ values and winter paleoclimate proxies”: what about a correlation with summer humidity, since this is that you say in the previous sentence was correlated in a previous study.

L.319-320: “The expansion of the LBK culture across central Europe took place during a dynamic climate period⁹, which may have influenced the evolution of foddering strategies and the use of leafy-hay.” Why? You need to explain the logic of this statement.

L.327-328: “Animals foraging under open canopied forests or at the edges of forests will have $\delta^{13}\text{C}$ values similar to those in open grassland.”: Reference?

L.335-336: “However, we cannot rule out that animals may have been grazing on herbaceous plants growing under an open forest canopy.” But you just said (L.327-328) that this would not result in low $\delta^{13}\text{C}$ values

L.344-360: “However, it is clear from this study and others⁵⁸ that detailed stable isotopic analysis of faunal material can provide detailed information about past environments.” I would argue that your findings actually show that it can't. You show no relationship between deer $\delta^{13}\text{C}$ values and MFC, which reflect the past environment.

L.364: “Previous analysis has shown that caprines had little access to forest resources³⁰ in comparison to cattle.” Except that previous determinations of sheep tooth AA- $\delta^{15}\text{N}$ values revealed consumption of woody plants?

L.365-367: “The lack of significant difference between $\delta^{13}\text{C}$ values from cattle skeletal material and dairy lipids supports the hypothesis that cattle were the primary source of milk for central European early farmers^{20,43}.” But how do the dairy lipid $\delta^{13}\text{C}$ values compare to those of caprines? You need to be able to exclude the alternative scenario, that they also derive from caprines.

L.372-373: “If leafy hay was used to promote lactation, it would explain the large variation in dairy lipids $\delta^{13}\text{C}$ values observed in some sites for example, at Cuiry-lès-Chaudardes.”: Why?

L.373-375: “Alternatively, the large variation in dairy lipid values may reflect also the inclusion of milk from caprines as suggested by the age-at-death profiles⁶⁰.” But this again contradicts your previous assertion that cattle milk was the primary dairy fat found in pottery vessels.

*****END*****

Print Email

Author Rebuttal, second revision:

Dr. Rosalind Gillis
Referat Naturwissenschaften
Im Dol 2-6
D 14195 Berlin

Tel.: +49 (0) 30 187711-308
rosalind.gillis@dainst.de
28.02.2024

Dear Editor and Reviewers,

We apologise for the long delay in sending the revised manuscript and we thank the reviewers for their patience as well as the time they have given to previous versions. We have made considerable changes in light of the comments made by the reviewers and the Editor and hope this revised version addresses the points raised in the last review.

The completely revised version of the manuscript has focused on:

1. Improving the narrative by describing how LBK farmers were experimental by nature by adapting crop cultivation methods to new environments, which begs the question whether animal husbandry was also adapted. We then describe the LBK cattle husbandry system detailing the changes in herd numbers seasonally to provide a frame of reference for the study (Figure 1A). Furthermore, we clarify the importance of animal feed on production, in particular milk production.
2. Clear description of the aims and analytical methods of the study in the introduction, supported by illustration of the 'canopy effect' and potential bioapatite signals that one would expect for

37different scenarios Figure 1B ('canopy effect') and Figure 1C (bioapatite signals). The methods section has also been revised to be avoid repetition with the main text.

3. To reduce contradictions highlighted by Reviewer 2 as well the disconnection between the results and discussion (Reviewer 1), we have now integrated the results and the discussions. This section is now composed of three main sections: stable carbon isotopes; seasonal diet and impact of climate on stable isotope proxies.
4. To improve clarity of the results, we have added Table 1: Summary of the datasets; Table 2: Correlation results as well as revising Figs 4 and 5 and their captions.
5. Previous comments from Reviewer 1, highlighted the lack of discussion about the results in general. Within Stable Carbon Isotopes, we discuss the distribution of the results from the three datasets (collagen, bioapatite and CSIA-lipids) and the variability in general and at individual sites.
6. We have separated reporting and discussion of our results concerning palaeo-environmental proxies: the forest cover proxy (MFC) within the sections exploring cattle diets (Stable Carbon Isotopes and Seasonal Diet) and then the palaeoclimatic data in the last section.
7. For the paleoclimatic data, we have highlighted the significant results namely that of the winter climate (PMW, TMW). However, there is a caveat to this result in that the climate proxies are an average for the duration of LBK culture in which there was strong variation in rainfall. Therefore we conclude it is not possible to conclusively state that the decreasing trend in carbon values is related to the wetter environments in the west compared to the east.
8. Via the revised Results/Discussion, we have clarified the arguments for our conclusions. These are that forests were used by the first farmers of central Europe for cattle grazing and fodder resources. The variation in practices demonstrates experimentation by communities with some areas being used more intensively than others. The use of forests for fodder and grazing would have a positive impact on cattle health and increasing milk production, while potentially having a negative impact on forests.

Yours sincerely,

Rosalind Gillis

38Detailed response to Reviewer 2 comments:

The authors seem to contradict themselves throughout the manuscript.

- The positive relationship between dairy lipid d13C values and MFC

In the revised text we have clearly stated, that this finding (which was both for the overall dataset and dairy lipid one) was contrary to expectations. We then suggest two explanations for this: 1. Use of forests where open areas are available i.e. MFC would be medium to low and d13C values would be low, an indication of forest fodder consumption (ln 233-237); 2. The problem with equifinality, where low d13C values could be a reflection of cattle consuming plants from waterlogged areas (ln 242-243).

- They maintain that low d13C values are due to consumption of woodland fodder, but then show using d15N-AA values that low d13C values are not consistently associated with consumption of lignin-rich plants.

This is completely expected as animals consuming herbaceous plants growing under a canopy will have negative values. We never observe consumption of woody plants and enriched d13C values.

The other assertions that they make in their abstract and conclusion – that “The long-term impact of forest pasture and fodder would have improved female cattle health, cattle breeding and milk availability for human consumption, as well as contributing to the emergence of man-modified forests.” derive from the results of other studies and not from this one.

Previous papers did not make the connection between forest fodders consumption and female health. This is a unique conclusion from our paper. In Balasse, Gillis et al. 2021, they discuss the practicalities involved with out-of-season births i.e. labour costs involved with intensive collection of supplementary fodders.

Clarification of research questions

Our main research focus is clearly stated in the abstract namely: To what extent did early farmers exploit forests to raise their herds? Therefore, we wish to explore the use of forests by LBK farmers to raise animals, which is a gap in our knowledge (ln 92-93) and is highly feasible given the other lines of evidence namely archaeobotany (ln 110-111). Alongside, how farmers adapted to the different forest landscapes encountered.

Conflation of too many factors (environment, climate and human behaviour).

Prior to describing human behaviour related to herding practices and thus evolution of these practices, it is important the environmental factors are assessed. In our revision, we have focused on the Mean Forest Cover proxy, as it is directly related to our research focus. Thus, we can describe the seasonality of foddering practices in light of the bioapatite analysis, which provides us some insight into potential human behaviours. For the climatic proxies, the correlation results and interpretation are discussed at the end of the results/discussion (Ln 324-338. Table 2).

There are still a considerable number of grammatical errors.

The paper has been completed revised and rigorously proof read.

L.69-70: “Our analysis reveals that farmers practiced a diverse set of pasturing strategies with the intensive use of forested ecosystems in some areas for both graze and seasonal forage reflecting the adaptation of herding to new environments”: Why does this indicate the adaptation to new environments? Wouldn’t it rather reflect a particular husbandry practice being practiced regardless of environment?

This has been changed (Ln 67-69: Our findings reveal a diversity of pasturing strategies for cattle employed by early farmers, with a notable emphasis on intensive utilisation of forests for grazing and seasonal foddering in some regions). At present, there is no evidence for the use of supplementary fodder from forests in SE Europe, prior to the emergence of the LBK communities. Therefore, it would appear that supplementing animal diets with leafy hay is an adaptation made by LBK farmers in response to local environments as well as a reflection of increasing consumption of milk products.

L.100: “waterlogged remains”: Of what? Cattle dung? Was it also sheep and goat dung?

This is has been changed (Ln122: waterlogged ruminant dung).

L.109: “photosynthesis”: What do you mean by this? The photosynthetic pathway?

Now Ln124-5, photosynthesis is the biological process that converts light energy into chemical energy and here we are talking about the efficiency of these processes under a canopy (Ln125-129).

L.120: “biosynthesis route for individual tissues”: ? Do you mean the different biosynthetic routing of nutrients into different tissues?

This has been removed.

L.126: “reflect are at an approximate...”: This doesn’t make grammatical sense and also will surely depend on the sampling density.

This has been removed.

L.139: “both sheep and cattle”: How many individuals?

This has been removed.

L.150: Are MFC predictions based on palynological data?

This has been updated (Ln155-6: mean forest cover (MFC)⁵⁷; palaeoclimate proxies derived from palynological data)

L.195: The distribution isn’t global. Rather overall range?

This has been changed (Ln187).

L.236: “If forest resources were used for cattle forage”: Surely, it is actually if the relative importance of forest resources for cattle forage was related to their availability? I.e. if there was more forest, cattle would have eaten more plants influenced by the canopy effect?

The reviewer is correct. However, we found that where there was greater availability of forests for grazing and supplementary feed (e.g. Ludwinowo), carbon values are higher than those areas where MFC was lower.

L.238: $r=-0.06$: This isn’t a correlation. It is basically showing that there is no relationship between MFC and $\delta^{13}C$ values.

This is a significant result albeit small.

L.245: “indication of cattle feeding in open water rich growing environments”: Why? What is the mechanism and its effect on plant $\delta^{13}C$ values?

We do not have the space to explain in detail why plants exhibit depleted $\delta^{13}C$ values. We reference to papers where interested readers can investigate further.

L.246-247: “Overall, the correlation analysis with the paleoenvironment proxies suggest that winter climatic conditions followed by MFC had the greatest influence on plant foliar $\delta^{13}C$ values”: This assertion is dubious given the differing correlations between the different proxies. You need to try and

explain the basis of this more convincingly.

As previously explained, the results/discussion section now focuses on the MFC proxy with a short discussion of correlation result between palaeoclimatic and $\delta^{13}\text{C}$ values.

L.250 onwards: This section provides much more convincing evidence for use of woody fodder and therefore use of forest resources. But it is based on a(n understandably) limited set of samples.

We cannot directly infer the consumption of supplementary feed collected from forests without compound specific analysis of amino acids from collagen and dentine samples. As with all palaeodietary studies of both humans and animals we infer the consumption of forest fodders based on $\delta^{13}\text{C}$ values. Unlike most palaeodietary studies that employ limited baselines, we use wild species as well as other domesticates (sheep) as a comparison as well as integrate carbon values from multi-tissues.

L.260: “potentially indicating different fodder types”: Or just access to plants growing in different levels of light intensity/water availability.

Yes we agree, and has been stated in Ln266-267.

L.287: instead of “within cattle fodder”, consumed by cattle. My understanding is that fodder is food deliberately collected for and given to livestock, rather than everything consumed by them, which could include grazed vegetation.

To prevent confusion we have limited the use of the word fodder. In general, we refer to animal diets, supplementary feed, and pasture.

L.296-298: Replace with: Figure 4A-F: Corrected combined stable isotopic results of incremental analysis of tooth enamel $\delta^{13}\text{C}$ values (grey diamonds, extended dashed grey line) and $\delta^{18}\text{O}$ values (white diamonds) and dentine AA- $\delta^{15}\text{N}$ values (red dotted line) from cattle teeth samples:

Now Figure 5. All figure captions have been revised.

Figure 4A-F: How do you model the red dotted line? Shouldn't you show the determined values rather than extrapolate? Or at least show both.

Now Figure 5. The $\delta^{18}\text{O}$ curve is the modelled curve based on the Balasse model for determining birth seasonality. The raw version of Figure 5 is found in Extended Figure 2.

L.318: “between diet $\delta^{13}\text{C}$ values and winter paleoclimate proxies”: what about a correlation with

summer humidity, since this is that you say in the previous sentence was correlated in a previous study.

This was the result of previous study where their paleoclimate values were based on modern values. We have reduced our discussion of palaeoclimatic values due to the reasons given above.

L.319-320: "The expansion of the LBK culture across central Europe took place during a dynamic climate period⁹, which may have influenced the evolution of foddering strategies and the use of leafy-hay." Why? You need to explain the logic of this statement.

This has been removed.

L.327-328: "Animals foraging under open canopied forests or at the edges of forests will have $\delta^{13}C$ values similar to those in open grassland.": Reference?

From our knowledge there has been no reference study that tests this. However, we have given two references that support this: Drucker et al. 2013; Tieszen 1991 (Ln140)

L.335-336: "However, we cannot rule out that animals may have been grazing on herbaceous plants growing under an open forest canopy." But you just said (L.327-328) that this would not result in low $d^{13}C$ values

This has been revised Ln278.

L.344-360: "However, it is clear from this study and others⁵⁸ that detailed stable isotopic analysis of faunal material can provide detailed information about past environments." I would argue that your findings actually show that it can't. You show no relationship between deer $d^{13}C$ values and MFC, which reflect the past environment.

This has been removed.

L.364: "Previous analysis has shown that caprines had little access to forest resources³⁰ in comparison to cattle." Except that previous determinations of sheep tooth AA- $d^{15}N$ values revealed consumption of woody plants?

From our knowledge there has been no AA-D15N analysis of sheep teeth.

L.365-367: "The lack of significant difference between $\delta^{13}C$ values from cattle skeletal material and dairy lipids supports the hypothesis that cattle were the primary source of milk for central European early farmers^{20,43}." But how do the dairy lipid $d^{13}C$ values compare to those of caprines? You need to be able

to exclude the alternative scenario, that they also derive from caprines.

There are three lines of evidence why cattle are the primary source of milk. 1. Lack of significant difference between cattle collagen $\delta^{13}\text{C}$ and dairy lipids values; 2. Kill-off profiles (Gillis et al. 2017); 3. Previous analysis of sheep and cattle collagen that showed that sheep had enriched $\delta^{13}\text{C}$ values (Zanon and Gillis 2020) (Ln197-202). We do not rule out that sheep may potential have been the source of dairy lipids.

L.372-373: "If leafy hay was used to promote lactation, it would explain the large variation in dairy lipids $\delta^{13}\text{C}$ values observed in some sites for example, at Cuiry-lès-Chaudardes.": Why?

The lactation season probably began in late spring and continued till autumn, during which the canopy of the forest changed. Therefore $\delta^{13}\text{C}$ of leafy hay may have varied during the season because of changes in the canopy, as well as mixing of milk sources (Ln309-311).

L.373-375: "Alternatively, the large variation in dairy lipid values may reflect also the inclusion of milk from caprines as suggested by the age-at-death profiles60." But this again contradicts your previous assertion that cattle milk was the primary dairy fat found in pottery vessels.

Please see the above comment.

Decision Letter, third revision:

Our ref: NATECOLEVOL-220315968C

10th April 2024

Dear Dr. Gillis,

Thank you for your patience as we've prepared the guidelines for final submission of your Nature Ecology & Evolution manuscript, "Diverse Prehistoric Cattle Husbandry Strategies in the Forests of Central Europe" (NATECOLEVOL-220315968C). Please carefully follow the step-by-step instructions provided in the attached file, and add a response in each row of the table to indicate the changes that you have made. Please also check and comment on any additional marked-up edits we have proposed within the text. Ensuring that each point is addressed will help to ensure that your revised manuscript

44can be swiftly handed over to our production team.

****We would like to start working on your revised paper, with all of the requested files and forms, as soon as possible (preferably within two weeks). Please get in contact with us immediately if you anticipate it taking more than two weeks to submit these revised files.****

In recognition of the time and expertise our reviewers provide to Nature Ecology & Evolution's editorial process, we would like to formally acknowledge their contribution to the external peer review of your manuscript entitled "Diverse Prehistoric Cattle Husbandry Strategies in the Forests of Central Europe". For those reviewers who give their assent, we will be publishing their names alongside the published article.

Nature Ecology & Evolution offers a Transparent Peer Review option for new original research manuscripts submitted after December 1st, 2019. As part of this initiative, we encourage our authors to support increased transparency into the peer review process by agreeing to have the reviewer comments, author rebuttal letters, and editorial decision letters published as a Supplementary item. When you submit your final files please clearly state in your cover letter whether or not you would like to participate in this initiative. Please note that failure to state your preference will result in delays in accepting your manuscript for publication.

Cover suggestions

We welcome submissions of artwork for consideration for our cover. For more information, please see our guide for cover artwork.

Nature Ecology & Evolution has now transitioned to a unified Rights Collection system which will allow our Author Services team to quickly and easily collect the rights and permissions required to publish your work. Approximately 10 days after your paper is formally accepted, you will receive an email in providing you with a link to complete the grant of rights. If your paper is eligible for Open Access, our Author Services team will also be in touch regarding any additional information that may be required to arrange payment for your article.

Please note that *Nature Ecology & Evolution* is a Transformative Journal (TJ). Authors may publish their research with us through the traditional subscription access route or make their paper immediately open access through payment of an article-processing charge (APC). Authors will not be required to make a final decision about access to their article until it has been accepted. Find out more about Transformative Journals

Authors may need to take specific actions to achieve compliance with funder and institutional open access mandates. If your research is supported by a funder that requires immediate open access (e.g. according to Plan S principles) then you should select the gold OA route, and we will direct you to the compliant route where possible. For authors selecting the subscription publication route, the journal's standard licensing terms will need to be accepted, including <https://www.nature.com/nature-portfolio/editorial-policies/self-archiving-and-license-to-publish>. Those licensing terms will supersede any other terms that the author or any third party may assert apply to any version of the manuscript.

[REDACTED]

[REDACTED]

Reviewer #1:

Remarks to the Author:

The authors have done substantial efforts to incorporate several of the points made by reviewers. As a result, the paper undoubtedly presents improvements, though there are still limitations, in particular regarding how uncertainty in proxies is tackled, and the interpretation of some of the correlation values. There is thus still a noticeable discrepancy between the robustness of the results, and the conclusion. Generally speaking, the authors are not very explicit about potential problems linked to the uncertainty of several of the proxies they use (e.g MFC which, as the name indicates, is merely a mean, but also paleoclimate data). Given that all measures used here rely upon interpolated maps which are then sampled at the point, this is problematic, as has further implications in the interpretations of their results.

The authors also make extensive use of correlation values, but often in somewhat hasty way. For instance, in their reply to comments by reviewer 2, they stress the importance of a r value of -0.06 , despite the fact that this clearly does not provide any robust statistical signal. Also, in the text, they highlight "most substantial relationships" between some of their isotopic signals with TMW and PMW, without pointing out that these two values are - without any surprise - themselves strongly correlated. In this sense, there is only one signal and the authors fail to realise this (checking autocorrelation of proxies should have been done beforehand, as, I think, I recommended in a previous round of comments). Also, the methods section is unclear as to whether or not a correction (Bonferonni or other) was applied, and how it impacts the reliability of the correlation values.

The results point to diverse regional strategies, which is an elegant but perhaps not so surprising result. Also, and especially in view of their aforementioned comments, it is noticeable that there is solid ground to consider foddering as a regional practice, but less so the direct use of forest pastures (see also the authors' own comments on equifinality (ie use of waterlogged environments). Yet, the paper ends with a paragraph entirely dedicated to human impact on forested environments, and also raises the idea of human management of the forests, for which the authors have effectively no strong data.

Reviewer #2:

Remarks to the Author:

I would like to thank the authors for their comprehensive revisions to the manuscript. I think that this

47paper is now much clearer and I am happy to recommend that it be published with minor changes (see below):

L.80: “domesticated cattle”?

L. 88: I think there are words missing in this sentence- it doesn’t make grammatical sense.

L. 92: remove the comma after “and”

L. 107: replace “later” with “latter”

L. 108: remove the comma after “and”

Fig. 1B: What is the reference(s) for the relationship between plant $\delta^{13}\text{C}$ values and canopy density?

L.129: replace “plants $\delta^{13}\text{C}$ discrimination” with: greater discrimination against $\delta^{13}\text{C}$.

L.145: I would replace “precise” with higher temporally resolved

L.150: Replace “CSIA- $\delta^{15}\text{N}$ ” with $\delta^{15}\text{N}_{\text{NAA}}$ values

L.186: I think you need to specify the datasets at the beginning of this section: so $\delta^{13}\text{C}_{\text{coll}}$, $\delta^{13}\text{C}_{\text{dairy}}$ and $\delta^{13}\text{C}_{\text{bioap}}$

L.202: replace “is” with being

L.206: I don’t think you mean above this “range”, because presumably you have provided the full range. Do you mean the mean value?

L.226: Do you mean “gallery forest” and “open deciduous woodland”?

L.227: replace “partly” with part

L.274: delete “and”

Fig. 4: Could you show the individual isotope values of the samples, overlaid on the violin plots?

Fig. 5: Could you show the individual $\delta^{18}\text{O}$ values that the sinusoidal curves are based upon?

L.318: replace “necessitated” with required

L.326: delete “a”

L.418-423: You don’t describe how methylation of the lipid extracts was carried out prior to analysis by GC-C-IRMS

L.52*; This sentence doesn’t make grammatical sense.

SI Table 5: Please provide all of the raw Beta values. For quality control you should also provide the $\delta^{15}\text{N}$ values of the standards and all amino acids in these dentine samples.

Author Rebuttal, second revision:

Dr. Rosalind Gillis
Abteilung Z I / Referat
Naturwissenschaften
Im Dol 2-6
D 14195 Berlin

Tel.: +49 (0) 30 187711-308
rosalind.gillis@dainst.de
07.05.2023

Dear Reviewers,

Thank you for your positive comments to our last submission. We believe that we have addressed all the concerns raised by both of you. We thank you for your patience and understanding during the long periods between revisions. Moreover, we appreciate the care and time that you have given, and as a result the comments and corrections have greatly improved the manuscript.

Please find below our point-by-point response to the reviewers.

Yours sincerely,

Rosalind Gillis

Reviewer #1 (Remarks to the Author):

Generally speaking, the authors are not very explicit about potential problems linked to the uncertainty of several of the proxies they use (e.g MFC which, as the name indicates, is merely a mean, but also paleoclimate data).

We agree with the reviewer there are problems with the proxies. However, we disagree that we do not consider this (see Ln269-78 for climate; Ln180-89 for MFC). Concerning MFC, this as with the climate data is a mean for the LBK period, this has been clarified in the methods section. Moreover, within the methods section, we describe why the MFC proxy was chosen and it should be acknowledged that this is the first time both paleo-climatic and vegetation proxies have been integrated into a stable isotope study.

49Given that all measures used here rely upon interpolated maps which are then sampled at the point, this is problematic, as has further implications in the interpretations of their results.

It is correct to state that interpolated models and sampling procedures are affected by uncertainties, and that these uncertainties in turn might influence further comparisons. It is also true that a degree of uncertainty is an intrinsic feature of all statistical models. It is important to point out that interpolation and point sampling are unavoidable steps when comparing geographically distributed, continental-scale datasets, as site- and time-specific environmental information is not available at the precise location where faunal data were collected. To mitigate these uncertainty-related issues, we make use of state-of-the-art data for both forest cover and palaeoclimate values covering the period under investigation. It is our opinion that assessing the uncertainties present within these datasets is beyond the scope of the manuscript.

The authors also make extensive use of correlation values, but often in somewhat hasty way. For instance, in their reply to comments by reviewer 2, they stress the importance of a r value of -0.06 , despite the fact that this clearly does not provide any robust statistical signal.

We agree this is not robust but it is a significant result. However, reporting of this result (MFC~Bioapatite) is not in the current MS that the reviewer commented on.

Also, in the text, they highlight "most substantial relationships" between some of their isotopic signals with TMW and PMW, without pointing out that these two values are - without any surprise - themselves strongly correlated. In this sense, there is only one signal and the authors fail to realise this (checking autocorrelation of proxies should have been done beforehand, as, I think, I recommended in a previous round of comments).

This has been corrected to refer to TMW/PMW as winter climate proxies (Ln272). We have added an Extended Figure 3, for visualisation of the autocorrelation between proxies and within the legend described the autocorrelation.

Also, the methods section is unclear as to whether or not a correction (Bonferonni or other) was applied, and how it impacts the reliability of the correlation values.

We have added Supplementary Table 8, which compares uncorrected p values with that of Bonferroni and also False Discovery Rate. The significant results discussed in the paper are highlighted in yellow. Based on this comparison our results are very reliable.

We have added this statement to the Methods: *Correlation tests were performed using the using the free platform R program with the Corrplot package (Supplementary Table 7). Uncorrected P values were compared with those corrected using Bonferroni correction (Supplementary Table 8). They were further*

assessed with the false discovery rate (FDR). This comparison showed that the correlation results are very reliable.

The results point to diverse regional strategies, which is an elegant but perhaps not so surprising result. Also, and especially in view of their aforementioned comments, it is noticeable that there is solid ground to consider foddering as a regional practice, but less so the direct use of forest pastures (see also the authors' own comments on equifinality (ie use of waterlogged environments). Yet, the paper ends with a paragraph entirely dedicated to human impact on forested environments, and also raises the idea of human management of the forests, for which the authors have effectively no strong data.

*We do not discuss the management of forests rather the modification of forest environments, which forest pasturing alongside other human activities would have caused. However, in light of the Reviewer's comment, we have adapted the last paragraph. To emphasize that we are talking about modification and not management in Ln309, the word *affected* has been replaced by *modified*. Moreover, we have removed the sentence: *At the same time, the intentional manipulation of forested landscapes underscores the dynamic interaction between early farmers and their environments, illuminating the transformative role of human agency in shaping the ecology and utilisation of forest resources.**

Final Decision Letter:

13th August 2024

Dear Dr Gillis,

We are pleased to inform you that your Article entitled "Diverse Prehistoric Cattle Husbandry Strategies in the Forests of Central Europe", has now been accepted for publication in Nature Ecology & Evolution.

Over the next few weeks, your paper will be copyedited to ensure that it conforms to Nature Ecology and Evolution style. Once your paper is typeset, you will receive an email with a link to choose the appropriate publishing options for your paper and our Author Services team will be in touch regarding any additional information that may be required

51Due to the importance of these deadlines, we ask you please us know now whether you will be difficult to contact over the next month. If this is the case, we ask you provide us with the contact information (email, phone and fax) of someone who will be able to check the proofs on your behalf, and who will be available to address any last-minute problems . Once your paper has been scheduled for online publication, the Nature press office will be in touch to confirm the details.

Acceptance of your manuscript is conditional on all authors' agreement with our publication policies (see www.nature.com/authors/policies/index.html). In particular your manuscript must not be published elsewhere and there must be no announcement of the work to any media outlet until the publication date (the day on which it is uploaded onto our web site).

Please note that *Nature Ecology & Evolution* is a Transformative Journal (TJ). Authors may publish their research with us through the traditional subscription access route or make their paper immediately open access through payment of an article-processing charge (APC). Authors will not be required to make a final decision about access to their article until it has been accepted. Find out more about Transformative Journals

Authors may need to take specific actions to achieve compliance with funder and institutional open access mandates. If your research is supported by a funder that requires immediate open access (e.g. according to Plan S principles) then you should select the gold OA route, and we will direct you to the compliant route where possible. For authors selecting the subscription publication route, the journal's standard licensing terms will need to be accepted, including <https://www.nature.com/nature-portfolio/editorial-policies/self-archiving-and-license-to-publish>. Those licensing terms will supersede any other terms that the author or any third party may assert apply to any version of the manuscript.

We welcome the submission of potential cover material (including a short caption of around 40 words) related to your manuscript; suggestions should be sent to Nature Ecology & Evolution as electronic files (the image should be 300 dpi at 210 x 297 mm in either TIFF or JPEG format). Please note that such pictures should be selected more for their aesthetic appeal than for their scientific content, and that colour images work better than black and white or grayscale images. Please do not try to design a cover with the Nature Ecology & Evolution logo etc., and please do not submit composites of images related to your work. I am sure you will understand that we cannot make any promise as to whether any of your suggestions might be selected for the cover of the journal.

You can generate the link yourself when you receive your article DOI by entering it here: <http://authors.springernature.com/share>.

Yours sincerely,

[REDACTED]

P.S. Click on the following link if you would like to recommend Nature Ecology & Evolution to your librarian <http://www.nature.com/subscriptions/recommend.html#forms>

** Visit the Springer Nature Editorial and Publishing website at www.springernature.com/editorial-and-publishing-jobs for more information about our career opportunities. If you have any questions please click here. **